# BRep: Graph-structured Brain Representation Learning via Parametric High-order Dependence Measures

## ABSTRACT

The brain network plays an important role in diagnosing neurological disorders. Brain functional network construction often follows the hand-crafted Correlation Coefficients of blood-oxygen-level-dependent (BOLD) time series without any learnable components. At the same time, most efforts are made to the models that predict individual neurological disorders with the constructed brain network as input, such as graph neural networks. Unfortunately, the fixed brain network may lose critical information during construction and lead to difficulty in performance improvement, even with deliberately designed graph models. From this perspective, the current situation is similar to the machine learning community, i.e., hand-crafted features and learnable predictors, before the advent of representation learning. In fact, the brain network can be regarded as a graph-structured learnable representation of the brain. By drawing on representation learning, this paper presents the Brain Representation (BRep) learning problem. To this end, the widely used linear and nonlinear correlations are enhanced to be high-order, parametric, and learnable. The flexible brain representation makes the following predictor simple and leads the framework to possess an end-to-end characteristic. The framework is implemented by combining the parametric correlation and a TopK sparsification. Extensive evaluations demonstrate that the proposed BRep possesses superior performance, high efficiency, and interpretability. The source code is publicly available at `https://anonymous.4open.science/r/BRep-demo/`

## 1 INTRODUCTION

The human brain represents the most intricate organ of the body. Research into its structure and function constitutes a central and invaluable endeavor in the life sciences, offering profound insights and serving as a critical guide for the diagnosis of neurological disorders . By characterizing the interactions among distinct brain regions, brain networks offer a novel framework and powerful tool for analyzing and advancing our understanding of the brain (Fornito et al., 2016).

The research on the brain network consists of three components: brain imaging (Tuan et al., 2025), brain network construction (Rubinov & Sporns, 2011), and brain network analysis (Bahrami et al., 2023). Brain imaging techniques can be broadly divided into structural imaging (sMRI (Fischl et al., 1999), DTI (Rubinov & Sporns, 2010), CT (He et al., 2007; Fleischer et al., 2019)), functional imaging (fMRI (Van Den Heuvel & Pol, 2010; Bielczyk et al., 2019), EEG (Rossini et al., 2019), MEG (Mandal et al., 2018), fNIRS (Li et al., 2022)), and molecular/metabolic imaging (PET (Veronese et al., 2019), SPECT (Imokawa et al., 2024), MRS (Soares & Law, 2009)). Brain network construction methods can be divided into functional connectivity (Van Den Heuvel & Pol, 2010), effective connectivity (Ray et al., 2021), and structural connectivity (Rubinov & Sporns, 2010). Brain network analysis often employs network/graph algorithms (Barabási, 2013; Foulds, 1995), including graph-theoretical measures (Rubinov & Sporns, 2010; Foulds, 1995), community algorithms (Newman, 2006), Graph Kernels (GKs) (Vishwanathan et al., 2010), Graph Neural Networks (GNNs) (Kipf, 2016), and Graph Transformers (GTs) (Dwivedi & Bresson, 2020; Kreuzer et al., 2021).

For the diagnosis of psychiatric and neurodegenerative disorders (Insel, 2010; Park & Friston, 2013), functional imaging (e.g., fMRI) and functional connectivity, which describes the statistical depen-

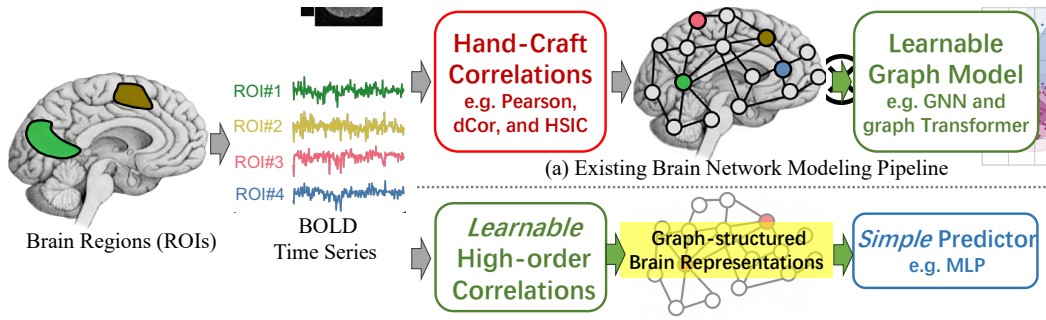

Figure 1: Comparison between the proposed graph-structured brain representation learning framework and the existing brain network modeling pipeline. They take the BOLD time series as input. (a) Existing modeling pipeline constructs a brain network with hand-crafted correlation coefficients without any learnable component. Then, the constructed brain network is fed into learnable flexible graph models. (b) The proposed brain representation learning framework considers the brain network as a flexible graph-structured brain representation. The representation is obtained via a learnable, high-order, parametric correlation estimator and make the downstream predictor simple.

dencies between brain regions, are widely used. Specifically, the brain network is constructed from the hand-crafted Correlation Coefficients, e.g., Pearson Correlation (Biswal et al., 1995), distance Correlation (Székely et al., 2007), and HSIC (Gretton et al., 2007), of BOLD (Ogawa et al., 1992). At the same time, most efforts are made on models that predict individual neurological disorders with the constructed brain network as input, and many flexible graph models, such as Graph Neural Networks (GNNs) (Li et al., 2021; Cui et al., 2022; Kan et al., 2022a; Zhang et al., 2023) and Graph Transformers (GTs) (Kan et al., 2022b; Xu et al., 2024; Yu et al., 2024; Peng et al., 2025), are explored as shown in Figure 1(a). Unfortunately, the fixed brain network may lose critical information during construction since simple and fixed correlation/dependence measures possess limited representation capability. Thus, it may lead to difficulty in performance improvement, even with deliberately designed graph models. The current situation is similar to the machine learning community, i.e., hand-crafted features and learnable predictors, before the advent of representation learning (Bengio et al., 2013; Domingos, 2012; LeCun et al., 2015).

By drawing on representation learning, this paper presents the Brain Representation (BRep) learning problem. In this problem, the brain network is regarded as a graph-structured learnable representation of the brain, instead of the fixed input to the graph models. The representation should be flexible to capture comprehensive information and make the following predictor simple, e.g., MLP, and accurate. To this end, the widely used linear and nonlinear correlations are enhanced to be high-order, parametric, and learnable. Specifically, linear and nonlinear correlations are unified as the inner product of the BOLD time series in high-dimensional latent spaces with *fixed* mappings, and the high-order dependence measure can be approximated with this inner product with a *learnable* mapping. The graph-structured representation and the parametric correlation estimator make the framework possess an end-to-end characteristic as shown in Figure 1(b). Finally, the framework is implemented by combining the parametric correlation and a TopK sparsification. The learning process is supervised by the individual's neurological disorder label and regularized by denoising the BOLD time series with the learned brain network. The main contributions of this paper are summarized as follows:

- We take the brain network as the representation of the brain, and present the problem of graph-structured brain representation learning.

- We propose a learnable, parametric high-order dependence measure by unifying and extending linear and nonlinear correlations.

- We implement a flexible brain representation learning method and simplify the downstream predictor to avoid iterative message passing operator.

- We experimentally demonstrate that the proposed brain representation learning is high-performance, scalable, and interpretable.

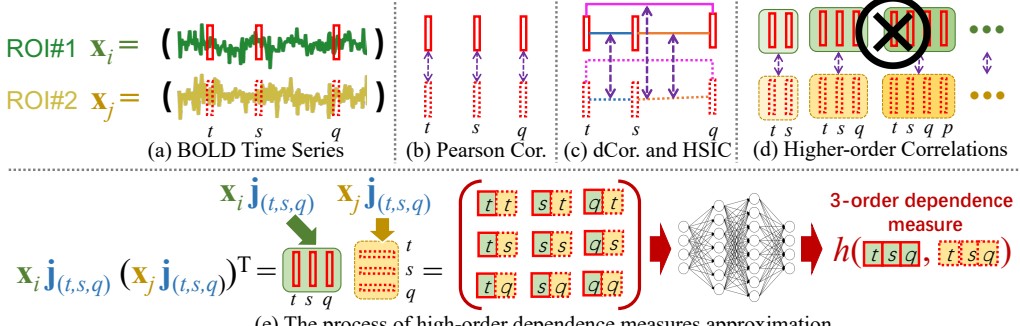

(a) BOLD Time Series  (b) Pearson Cor.  (c) dCor. and HSIC  (d) Higher-order Correlations

(e) The process of high-order dependence measures approximation

Figure 2: Comparison between different correlation estimators. (a) The BOLD time series of two brain regions (ROIs). The representative samples are indexed with $t$, $s$, and $q$. (b) Pearson Correlation estimates linear correlation between two samples in each index, respectively. (c) dCor and HSIC estimate nonlinear correlations between relationships (distance in dCor and kernel in HSIC) in pairs of indexes. Links between samples with different colors represent different index pairs. (d) High-order Dependence measure extends to the combination of any subsets of indexes. (e) An example of high-order dependence measure.

## 2 GRAPH-STRUCTURED BRAIN REPRESENTATION LEARNING

As shown in Fig. 1(a), existing brain network modeling methods construct a brain network by employing different correlation coefficients, which are comprehensively developed in statistics without a learnable component. Flexible graph models, such as GNNs and GTs, are trained by feeding the constructed brain network as input. Unfortunately, the fixed brain network may lose critical information during construction and prevent performance improvement in the following graph model, since simple dependence measures possess limited representation capability.

A good representation should make downstream tasks easier, and thus, the predictor should be simple. To this end, the representation should be flexible to capture comprehensive information. For the brain analysis task, which takes BOLD time series as input, it is challenging to directly obtain a representation in vector form. To alleviate this difficulty, this paper seeks a graph-structured brain representation instead of the one in vector form. To this end, a learnable, high-order, parametric correlation estimator is proposed as shown in Fig. 1(b). This framework possesses the end-to-end characteristic. This section begins by providing notations and a problem definition (Section 2.1). Then, a high-order dependence measure with learnable parameters is introduced, and its connection to existing linear and nonlinear correlation coefficients is established (Section 2.2). Finally, an end-to-end interpretation is proposed (Section 2.3).

### 2.1 NOTATIONS AND PROBLEM DEFINITION

The brain is divided into $N$ ROIs, i.e., $\mathcal{V} = \{v_1, v_2, ..., v_N\}$. The BOLD time series of the $i$-th ROI is represented by $\mathbf{x}_i = (x_{i1}, x_{i2}, \cdot, x_{iD}) \in \mathbb{R}^{1 \times D}$, where $D$ is the length of the time series. $\mathbf{X} \in \mathbb{R}^{N \times D}$ is the collection of BOLD time series of $N$ brain regions with $\mathbf{x}_i$ as $i$-th row. Brain network, which models the connectivity between ROIs, can be denoted as a graph $G = (\mathcal{V}, \mathcal{E})$, where $\mathcal{E}$ denotes the collection of edges between nodes/ROIs. The graph topology is represented as the adjacency matrix $\mathbf{A} = [a_{ij}] \in \mathbb{R}^{N \times N}$ is where $a_{i,j}$ is the weights between nodes $v_i$ and $v_j$. $\mathcal{N}(v)$ denotes the neighbourhoods of node $v$.

For brain analysis tasks, a set of $L$ subjects' brain data $\mathcal{X} = \{\mathbf{X}^{(1)} \ldots \mathbf{X}^{(L)}\}$ and the corresponding labels $\mathcal{Y} = \{y^{(1)} \ldots y^{(L)}\}$, which indicates biological sex, presence of a disease or other properties of the brain subject, are provided. Brain network modeling aims to learn from given data $\mathcal{X}$ and $\mathcal{Y}$ by designing a function $y = f(h(\mathbf{X}))$, which is composed of a brain construction function $G = h(\mathbf{X})$ and a prediction function $y = f(G)$ based on the constructed brain network $G$. Existing brain network modeling methods employ a fixed brain construction function $h(\cdot)$ and train a prediction function $f(\cdot)$, while the proposed graph-structured brain representation learning framework jointly trains $h(\cdot)$ and $f(\cdot)$ in an end-to-end manner.

## 2.2 From Linear/Nonlinear Correlations to High-order Dependence Measures

This section first reviews representative linear and nonlinear correlations. Then, high-order dependence measures with learnable parameters are derived by unifying the linear and nonlinear ones.

### 2.2.1 Liner Correlation: Pearson Correlation

Pearson Correlation Coefficient is a widely used linear correlation measure, and is the default in brain network construction. The Pearson Correlation Coefficient between $\mathbf{x}_i$ and $\mathbf{x}_j$ is defined as:

$$r_{ij} = \frac{\text{Cov}(\mathbf{x}_i, \mathbf{x}_j)}{\sigma_{\mathbf{x}_i} \sigma_{\mathbf{x}_j}} = \frac{\sum_t (x_{it} - \bar{x}_i)(x_{jt} - \bar{x}_j)}{\sqrt{\sum_t (x_{it} - \bar{x}_i)^2 \sum_t (x_{jt} - \bar{x}_j)^2}},$$

where $\bar{x}_i$ and $\sigma_{\mathbf{x}_i}$ are the mean and standard deviation of $\mathbf{x}_i$, respectively. If elements in $\mathbf{x}_i$ is normalized as $\tilde{x}_{it} = (x_{it} - \bar{x}_i)/\sigma_{\mathbf{x}_i}$, $r_{ij}$ can be simplified as

$$r_{ij}^{[1]} = \sum_t \tilde{x}_{it} \tilde{x}_{jt} = \tilde{\mathbf{x}}_i \tilde{\mathbf{x}}_j^T = (\tilde{\mathbf{x}}_i \mathbf{I}) (\tilde{\mathbf{x}}_j \mathbf{I})^T, \tag{1}$$

where row vector $\tilde{\mathbf{x}}_i = (\tilde{x}_{i1}, \tilde{x}_{i2}, \cdot, \tilde{x}_{iD})$ and $\mathbf{I}$ is the identity matrix. As shown in Figure. 2(b), it estimates the correlation between two samples in each index, e.g. $x_{it}$ and $x_{jt}$ with index $t$ in Eq. (1). Although the Pearson Correlation Coefficient is computationally efficient, the characteristic of capturing linear correlation limits its capability.

### 2.2.2 Nonlinear Correlation: Distance Correlation and HSIC

To overcome the limitation of the linear one, the nonlinear correlations are proposed to capture nonlinear dependences. Instead of using two samples in each index in linear correlations, it employs two distances between samples with the same index pair, e.g. $dis(x_{it}, x_{is})$ and $dis(x_{jt}, x_{js})$ with index pair $(t, s)$ as shown in Fig. 2(c). If two random variables are independent, the covariance between pairs of sample distances approaches zero. Two representative instances of nonlinear correlation are distance Correlation (dCor) and Hilbert-Schmidt Independence Criterion (HSIC), which share a similar formula as

$$r_{ij} = \frac{1}{D^2} \text{tr}(\mathbf{AB}) = \frac{1}{D^2} \sum_{t,s} A_{ts} B_{ts}, \tag{2}$$

where $\mathbf{A} = (A_{ts})_{D \times D}$ and $\mathbf{B} = (B_{ts})_{D \times D}$ are the distance matrices with $A_{ts}$ and $B_{ts}$ denoting the distances of sample pairs, i.e., $dis(x_{it}, x_{is})$ and $dis(x_{jt}, x_{js})$, respectively. $\text{tr}(\cdot)$ represents the trace of the matrix. dCor defines the distance in European space, i.e., $dis(x, y) = |x - y|$, while HSIC employs the kernel trick and defines the distance in RKHS with the Gram matrix (Schölkopf & Smola, 2002), i.e., $dis(x, y) = \text{kernel}(x, y)$. For simplicity, Eq. (2) can be rewritten as $r_{ij} = \frac{1}{D^2} \sum_{t,s} |x_{it} - x_{is}| \cdot |x_{jt} - x_{js}|$. By ignoring the absolute value constraint and normalization $1/D^2$, it can be formulated as

$$r_{ij}^{[2]} = \sum_{t,s} (x_{it} - x_{is}) \cdot (x_{jt} - x_{js}) = (\mathbf{x}_i \mathbf{J})(\mathbf{x}_j \mathbf{J})^T = \mathbf{x}_i (\mathbf{JJ}^T) \mathbf{x}_i^T, \tag{3}$$

where $\mathbf{J}$ is to compute the difference of a pair of elements. $\mathbf{J} \in \mathbb{R}^{D \times D^2}$ contain $D^2$ columns, each of which corresponds to one index pair $(t, s)$ and possesses the following form

$$\mathbf{j}_{(t,s)} = [0, \cdots, 0, \underset{t}{1}, 0, \cdots, 0, \underset{s}{-1}, 0, \cdots, 0]^T, \tag{4}$$

where $t$-th and $s$-th elements are $1$ and $-1$, respectively, while all other elements are $0$. dCor and HSIC compromise between the efficiency and capability of capturing nonlinear correlation. Therefore, nonlinear correlation possesses limited capability in capturing high-order dependencies.

### 2.2.3 High-order Dependence Measures

To alleviate the limitation in nonlinear correlation, high-order dependence should employ two sample tuples indexed by $t_1, t_2, \cdots, t_M$ as shown in Fig. 2(d) (Hoeffding, 1992; Lee, 2019; Sejdinovic et al., 2013). Motivated by U/V-statistics, its formula can be expressed as

$$r_{ij} = \frac{1}{\binom{D}{M}} \sum_{t_1, t_2, \cdots, t_M} f\big((x_{it_1}, x_{it_2}, \cdots, x_{it_M}), (x_{jt_1}, x_{jt_2}, \cdots, x_{jt_M})\big), \tag{5}$$

where $f(\cdot, \cdot)$ denotes the product of distance, kernel, or other measures. Tensor-HSIC implements $f(\cdot, \cdot)$ with a tensor product kernel, which causes high computational complexity ($\mathcal{O}(D^3)$).

Note that linear correlation in Eq. (1) and approximate nonlinear correlation in Eq. (3) have a similar form. In Eq. (1), the identity matrix $\mathbf{I}$ has $D$ columns, each of which possesses one nonzero element. In Eq. (3), the difference matrix $\mathbf{J}$ has $D^2$ columns, each of which possesses two nonzero elements. Thus, Eq. (3) can also be used to approximate correlation with $M$ samples by extending $\mathbf{J}$ with $D^M$ columns and letting each column have $M$ nonzero elements, which corresponds to the index tuple $t_1, t_2, \cdots, t_M$. Unfortunately, it also leads to high computational complexity.

However, the rank of $\mathbf{J}$ in Eq. (3) is at most $D$, no matter how many columns it has. Thus, the rank of the product $\mathbf{J}\mathbf{J}^T$ is at most $D$. Therefore, instead of seeking $\mathbf{J}$, the matrix $\mathbf{J}\mathbf{J}^T$ can be approximated by the product of a square matrix and its transpose, i.e., $\mathbf{J}\mathbf{J}^T = \mathbf{O}\mathbf{O}^T$ where $\mathbf{O} \in \mathbb{R}^{D \times D}$, and Eq. (3) can be reformulated to

$$r_{ij}^{[high]} = \mathbf{x}_i \left( \mathbf{O}\mathbf{O}^T \right) \mathbf{x}_i^T = \left( \mathbf{x}_i \mathbf{O} \right) \left( \mathbf{x}_j \mathbf{O} \right)^T, \tag{6}$$

where $\mathbf{O} \in \mathbb{R}^{D \times D}$ contains the parameters of this high-order dependence estimator. Correlation parameter $\mathbf{O}$ as well as the parameters of the predictor can be jointly trained in an end-to-end manner, as shown in Fig. 1(b). Note that the requirement of the number of columns of $\mathbf{O}$ being $D$ is critical for performance. On one hand, if the number of columns of $\mathbf{O}$ is less than $D$, it may be underfitting to $\mathbf{J}\mathbf{J}^T$ with rank $D$. On the other hand, if the number of columns of $\mathbf{O}$ is much larger than $D$, it may be overfitting due to the large number of parameters. Furthermore, Eq. (6) can also be used to approximate the combination of multiple correlations with different orders as shown in Fig. 2(d).

### 2.3 END-TO-END IMPLEMENTATION

Based on the parametric high-order dependence estimator in Eq. (6), a scalable implementation is presented in this section. The graph-structured brain representation $\mathbf{A}$ is obtained as follows:

$$\mathbf{A} = \sigma \Big( \text{Norm} \big( \text{TopK}(\mathbf{Z}\mathbf{Z}^T) \big) \Big), \quad \mathbf{Z} = \hat{\mathbf{X}}\mathbf{O}, \tag{7}$$

where $\text{TopK}(\cdot)$ selects top $K$ elements in each row, $\text{Norm}(\cdot)$ is row-wise normalization, $\sigma(\cdot)$ denotes nonlinear activation function, $\hat{\mathbf{X}}$ is the row-wise z-score normalized version of $\mathbf{X}$, and $\mathbf{X}, \hat{\mathbf{X}}, \mathbf{Z} \in \mathbb{R}^{N \times D}, \mathbf{A} \in \mathbb{R}^{N \times N}, \mathbf{O} \in \mathbb{R}^{D \times D}$. This component is designated as the **Mapping of High-order Dependence Measure (HDM)**. With the flexible graph-structured brain representation $\mathbf{A}$ in Eq. (7), the following predictor can be simple. Here, multi-layer perceptron (MLP) and pooling function are employed, i.e.,

$$\hat{y} = f_{MLP} \Big( \text{pooling} \big( f_{MLP}(\mathbf{A}) \big) \Big).$$

The parameters of correlation and $f_{MLP}$'s can be jointly trained with the supervision of the cross-entropy, i.e., $\mathcal{L}_{CE} = \sum_{l=1}^{L} \text{cross} - \text{entropy}(\hat{y}^{(l)}, y^{(l)})$, between the ground-truth $y$ and predicted label $\hat{y}$. To stabilize the training process, a regularization term, derived from a denoising autoencoder (DAE), is introduced to the objective function by denoising the corrupted BOLD time series with the learned brain representation $\mathbf{A}$, i.e.,

$$\mathcal{L} = \mathcal{L}_{CE} + \lambda \mathcal{L}_{reg}, \quad \mathcal{L}_{reg} = \big\| \mathbf{X} - \text{GNN}(\tilde{\mathbf{X}}, \mathbf{A}) \big\|,$$

where $\lambda$ is the hyper-parameter to balance the cross-entropy and regularization, $\tilde{\mathbf{X}}$ is the corrupted version of $\mathbf{X}$, and $\text{GNN}(\cdot, \cdot)$ is GCN to denoise the corrupted $\tilde{\mathbf{X}}$ using the learned brain network $\mathbf{A}$.

To better understand what the high-order correlation dependency measure is, an example of third-order dependency measure is shown in Fig. 2 (e). Three parameters $t, s, q$ in HDM can be combined through matrix product to form matrix $\mathbf{A}$, within which all pairwise interactions among the three component matrices are implicitly encoded. The matrix $A$, containing all pairwise relationships among $t, s, q$, can be fed into $MLP$, which is capable of composing these interactions into arbitrary third-order dependence measure.

## 2.4 THEORETICAL ANALYSIS

This section provides a theoretical analysis to show that the proposed bilinear function in Eq. (6) can approximate any U/V-statistics in Eq. (5). To this end, the high-order U/V-statistics can be regarded as a continuous function. Specifically, $m$-order U/V-statistics can be defined as $h : K \to \mathbb{R}$ where $K \subset \mathbb{R}^{2m}$ be a compact set. Denoting by $\mathcal{H}$ the class of feedforward neural networks (MLPs) with a non-polynomial activation (e.g. ReLU, sigmoid or tanh), the universal approximation theorem (UAT) (Hornik et al., 1989; Sonoda & Murata, 2017) shows that for every compact set $C$ and continuous $f : C \to \mathbb{R}$ there exists a network $g \in \mathcal{H}$ approximating $f$ uniformly on $C$ to arbitrary precision. The theoretical analysis is explored via the following three steps.

- Explicitly forming bilinear features (outer-product entries) after a global invertible linear mixing and feeding them into an expressive MLP yields a class of architectures that is universal for continuous high-order dependence mappings on compact domains.

- Reducing the relation between two sample groups to a single scalar inner-product and applying a single-variable nonlinearity is strictly limited in expressivity and cannot approximate arbitrary continuous high-order dependence functions.

- Using multiple inner-product channels (multi-head inner-products) can reconstruct the full outer-product when the number of channels reaches $m^2$ and the projections are chosen appropriately; hence, multi-head constructions can recover universality.

The following theorem provides the universality of the bilinear function with the outer-product.

**Theorem 2.1.** *Let $K \subset \mathbb{R}^{2m}$ be compact and $h \in C(K)$. Let $\mathbf{W} \in \mathbb{R}^{m \times m}$ be invertible and define*

$$T : K \to \mathbb{R}^{2m+m^2}, \qquad T(\mathbf{x}, \mathbf{y}) = (\mathbf{u}, \mathbf{v}, \text{vec}(\mathbf{u}\mathbf{v}^\top)), \qquad \mathbf{u} = \mathbf{W}\mathbf{x}, \ \mathbf{v} = \mathbf{W}\mathbf{y}.$$

*Then for every $\varepsilon > 0$ there exists a neural network $g \in \mathcal{H}$ (with input dimension $2m + m^2$) such that*

$$\sup_{(\mathbf{x}, \mathbf{y}) \in K} \left| h(\mathbf{x}, \mathbf{y}) - g(T(\mathbf{x}, \mathbf{y})) \right| < \varepsilon.$$

*In other words, the class $\{(\mathbf{x}, \mathbf{y}) \mapsto g(T(\mathbf{x}, \mathbf{y})) : g \in \mathcal{H}\}$ is dense in $C(K)$.*

The proof is provided in Appendix F.1.

**Remark 2.2.** The invertibility of $\mathbf{W}$ is a sufficient condition that guarantees injectivity of the intermediate linear mixing; it ensures no information about $(\mathbf{x}, \mathbf{y})$ is lost before forming bilinear features. If $\mathbf{W}$ is rank-deficient, the mapping $T$ may collapse distinct $(\mathbf{x}, \mathbf{y})$ to the same $T(\mathbf{x}, \mathbf{y})$; then only those target functions constant on fibers of $T$ can be represented as $g \circ T$.

However, the inner-product bilinear function, i.e., bilinear function without the outer-product, can not universally approximate continuous functions. To this end, the single-scalar pipeline function class is defined as follows, and theoretical analysis is given in Theorem 2.4.

**Definition 2.3.** Define the single-scalar pipeline function class

$$\mathcal{F}_1 := \{(\mathbf{x}, \mathbf{y}) \mapsto \phi(\mathbf{x}^\top M \mathbf{y}) : M \in \mathbb{R}^{m \times m}, \ \phi \in C(\mathbb{R})\}.$$

**Theorem 2.4.** *Let $m \geq 2$ and let $K \subset \mathbb{R}^{2m}$ be any compact set that contains a set of the form $\{(\mathbf{x}, \mathbf{y}_0) : \mathbf{x} \in U\}$ where $U \subset \mathbb{R}^m$ contains two points $\mathbf{x}^{(1)} \neq \mathbf{x}^{(2)}$ with $\mathbf{x}^{(1)} - \mathbf{x}^{(2)} \notin \ker(M\mathbf{y}_0)$ for every $M$ in a prescribed collection (in particular one may take $U$ to have nonempty interior). Then there exists a continuous function $h \in C(K)$ and a constant $\delta > 0$ such that for every $f \in \mathcal{F}_1$,*

$$\|h - f\|_{L^\infty(K)} \geq \delta.$$

*Consequently, $\mathcal{F}_1$ is not dense in $C(K)$.*

The proof is provided in Appendix F.2.

**Remark 2.5.** The intuitive reason is that for fixed $\mathbf{y}$, each $f \in \mathcal{F}_1$ reduces to a single-variable function of a one-dimensional linear projection of $\mathbf{x}$, while generic continuous functions on $\mathbf{x}$ cannot be represented through a single linear projection and a scalar nonlinearity.

Table 1: Performance comparison with three types of baselines, reported as percentages ($mean_{\pm std}$) over five trials. Best and runner-up models are highlighted in red and blue, respectively. *GNNs*, *GTs*, and *NNs* stand for graph neural networks, graph transformers, and neural networks, respectively.

| Type | Model | ABIDE | | | | ADHD-200 | | | |
|------|-------|-------|-------|-------|-------|----------|-------|-------|-------|
| | | AUC ↑ | ACC ↑ | SEN ↑ | SPE ↑ | AUC ↑ | ACC ↑ | SEN ↑ | SPE ↑ |
| *GNNs* | GCN | $59.59_{\pm 3.44}$ | $59.30_{\pm 3.38}$ | $56.67_{\pm 4.37}$ | $61.55_{\pm 5.29}$ | $67.01_{\pm 3.56}$ | $64.92_{\pm 6.32}$ | $65.09_{\pm 6.03}$ | $62.24_{\pm 5.90}$ |
| | GAT | $60.43_{\pm 3.88}$ | $60.10_{\pm 4.13}$ | $59.26_{\pm 5.51}$ | $62.89_{\pm 8.03}$ | $61.97_{\pm 3.28}$ | $63.38_{\pm 3.18}$ | $64.54_{\pm 13.97}$ | $45.10_{\pm 18.34}$ |
| | BrainGNN | $64.42_{\pm 3.57}$ | $63.09_{\pm 1.35}$ | $65.65_{\pm 2.88}$ | $60.67_{\pm 3.68}$ | $67.19_{\pm 2.86}$ | $65.16_{\pm 3.81}$ | $65.09_{\pm 2.11}$ | $64.43_{\pm 3.81}$ |
| | BrainGB | $70.32_{\pm 3.66}$ | $65.12_{\pm 3.90}$ | $67.01_{\pm 10.00}$ | $60.07_{\pm 8.53}$ | $75.23_{\pm 11.02}$ | $69.34_{\pm 7.41}$ | $67.46_{\pm 9.82}$ | $68.15_{\pm 8.41}$ |
| | FBNETGEN | $74.55_{\pm 3.77}$ | $67.09_{\pm 3.37}$ | $64.71_{\pm 9.85}$ | $69.61_{\pm 9.30}$ | $77.40_{\pm 4.76}$ | $68.82_{\pm 6.27}$ | $66.45_{\pm 7.73}$ | $71.56_{\pm 14.07}$ |
| | A-GCL | $73.86_{\pm 2.91}$ | $71.04_{\pm 2.40}$ | $71.42_{\pm 3.03}$ | $70.95_{\pm 3.19}$ | $74.78_{\pm 4.39}$ | $73.11_{\pm 4.30}$ | $72.04_{\pm 4.68}$ | $73.08_{\pm 4.10}$ |
| *GTs* | SAN | $71.35_{\pm 2.18}$ | $65.34_{\pm 2.91}$ | $55.41_{\pm 9.29}$ | $68.39_{\pm 7.50}$ | $51.22_{\pm 2.21}$ | $51.09_{\pm 2.00}$ | $50.43_{\pm 19.32}$ | $51.74_{\pm 20.16}$ |
| | Graphormer | $63.91_{\pm 4.05}$ | $61.88_{\pm 6.85}$ | $66.30_{\pm 9.98}$ | $55.74_{\pm 11.00}$ | $58.64_{\pm 1.50}$ | $61.60_{\pm 0.90}$ | $83.34_{\pm 2.90}$ | $33.96_{\pm 6.10}$ |
| | GraphTrans | $60.13_{\pm 6.73}$ | $57.83_{\pm 4.71}$ | $65.70_{\pm 10.30}$ | $49.77_{\pm 11.52}$ | $51.49_{\pm 1.15}$ | $50.76_{\pm 2.07}$ | $62.39_{\pm 9.43}$ | $39.13_{\pm 10.74}$ |
| | BrainNETTF | $77.93_{\pm 1.41}$ | $69.26_{\pm 2.26}$ | $65.92_{\pm 8.60}$ | $73.20_{\pm 6.06}$ | $79.79_{\pm 3.14}$ | $72.67_{\pm 3.17}$ | $73.64_{\pm 11.06}$ | $72.08_{\pm 5.66}$ |
| | ContrastPool | $57.36_{\pm 0.87}$ | $57.44_{\pm 0.69}$ | $57.66_{\pm 6.85}$ | $57.08_{\pm 7.79}$ | $71.19_{\pm 2.26}$ | $69.16_{\pm 2.85}$ | $67.71_{\pm 3.15}$ | $70.59_{\pm 2.91}$ |
| | ALTER | $77.99_{\pm 2.21}$ | $70.10_{\pm 2.26}$ | $72.84_{\pm 7.40}$ | $67.68_{\pm 5.81}$ | $83.16_{\pm 1.61}$ | $73.48_{\pm 1.38}$ | $74.58_{\pm 6.85}$ | $72.20_{\pm 5.82}$ |
| | BioBGT | $69.96_{\pm 1.18}$ | $69.70_{\pm 2.92}$ | $67.04_{\pm 3.41}$ | $72.02_{\pm 4.67}$ | $71.64_{\pm 1.14}$ | $71.06_{\pm 0.08}$ | $75.39_{\pm 5.45}$ | $71.92_{\pm 2.29}$ |
| *NNs* | MLP | $75.60_{\pm 2.38}$ | $70.92_{\pm 2.34}$ | $63.96_{\pm 9.58}$ | $73.03_{\pm 7.68}$ | $78.36_{\pm 1.88}$ | $70.68_{\pm 4.98}$ | $74.15_{\pm 4.63}$ | $64.42_{\pm 7.60}$ |
| | BQN | $79.85_{\pm 1.27}$ | $72.53_{\pm 1.41}$ | $73.26_{\pm 5.99}$ | $72.03_{\pm 6.24}$ | $83.34_{\pm 1.13}$ | $75.68_{\pm 1.95}$ | $79.73_{\pm 2.27}$ | $71.63_{\pm 4.87}$ |
| *Ours* | BRep | $77.64_{\pm 2.06}$ | $73.58_{\pm 3.67}$ | $74.86_{\pm 8.16}$ | $68.12_{\pm 5.21}$ | $84.53_{\pm 2.85}$ | $77.82_{\pm 5.50}$ | $78.11_{\pm 3.12}$ | $73.30_{\pm 4.90}$ |

To alleviate the limitation inner-product bilinear function on universal approximation, *multi-head inner-products* are proposed to approximate the outer-product. The following theorem provides the approximation error.

**Theorem 2.6.** *Let $m \geq 1$ and let $K \subset \mathbb{R}^{2m}$ be compact. For $R = m^2$ define matrices $\{\mathbf{U}^{(i,j)}, \mathbf{V}^{(i,j)}\}_{1 \leq i,j \leq m}$ by*

$$\mathbf{U}^{(i,j)} := \mathbf{e}_i^\top \in \mathbb{R}^{1 \times m}, \qquad \mathbf{V}^{(i,j)} := \mathbf{e}_j^\top \in \mathbb{R}^{1 \times m},$$

*where $\mathbf{e}_k$ is the $k$-th standard basis column vector in $\mathbb{R}^m$. Define scalar channels*

$$s_{ij}(\mathbf{x}, \mathbf{y}) := (\mathbf{U}^{(i,j)} \mathbf{x})^\top (\mathbf{V}^{(i,j)} \mathbf{y}) = \mathbf{x}_i \mathbf{y}_j.$$

*Let $s(\mathbf{x}, \mathbf{y}) \in \mathbb{R}^{m^2}$ be the vector obtained by ordering $\{s_{ij}\}_{i,j}$. Then $s(\mathbf{x}, \mathbf{y}) = \mathrm{vec}(\mathbf{x}\mathbf{y}^\top)$ and the mapping $(\mathbf{x}, \mathbf{y}) \mapsto s(\mathbf{x}, \mathbf{y})$ is continuous and injective on any set where $(\mathbf{x}, \mathbf{y}) \mapsto \mathbf{x}\mathbf{y}^\top$ is injective. Consequently, for any continuous $h : K \to \mathbb{R}$ and any $\varepsilon > 0$ there exists an MLP $g \in \mathcal{H}$ such that*

$$\sup_{(\mathbf{x},\mathbf{y}) \in K} \left| h(\mathbf{x}, \mathbf{y}) - g(s(\mathbf{x}, \mathbf{y})) \right| < \varepsilon.$$

The proof is provided in Appendix F.3.

**Remark 2.7.** The matrices chosen in the constructive proof are the simplest possible: each channel extracts exactly one pairwise product $\mathbf{x}_i \mathbf{y}_j$. In practice one often uses fewer channels $R < m^2$ with learned or random projection matrices $\mathbf{U}^{(r)}, \mathbf{V}^{(r)}$ and relies on the downstream MLP to recover or approximate the necessary combinations.

By combining Theorems 2.1 and 2.6, the bilinear function with multi-head inner-products possesses the property of universal approximation.

## 3 EXPERIMENT

This section provides a comprehensive evaluation of the proposed BRep. The experimental setup is described in Appendix B. Quantitative performance comparisons with representative baselines are subsequently reported and the key findings are discussed. Finally, additional analyses are provided, including ablation studies, interpretability analyses, timeseries comparisons, and hyperparameter sensitivity.

### 3.1 EXPERIMENTAL RESULTS

**Brain Disorder Disease Classification.** Tab. 1 reports the performance comparison between the proposed BRep and representative baselines on the ABIDE and ADHD-200. Overall, the proposed

models BRep consistently achieve competitive performance, outperforming most baselines in terms of AUC, ACC, SEN, and SPE. Therefore, two conclusions can be drawn. (1) The performance of BRep depends on the quality of the input brain network, not only on advanced architectural design. Although GNNs and GTs-based models have attained promising performance by relying on sophisticated architectural designs, the proposed BRep, based on learnable brain network construction and simple NNs, achieves superior results. For example, on the ADHD-200 dataset, the proposed BRep shows the best results on nearly all metrics. Particularly, the models outperform the second-best BQN by $2.14\%$ on ACC. (2) The proposed **HDM** module is effective. It can be observed that, compared to the baselines MLP and BQN, which rely on the hand-crafted Pearson Correlation Coefficient, the proposed BRep achieves performance improvements on most metrics. The results highlight the high quality of the constructed brain networks, thereby confirming the effectiveness of the construction module.

## 3.2 INTERPRETABILITY ANALYSIS

**Case Study.** This experiment aims to assess the biological interpretability of the proposed BRep by visualizing differential brain networks, *i.e.*, connectivity differences between Normal Controls (NC) and Autism Spectrum Disorder (ASD) groups. In the first step, group-level connectivity templates are calculated by averaging the functional connectivity matrices within each group: $A_{\text{Template}}^{\text{ASD}} = \frac{1}{n_1} \sum_{i=1}^{n_1} A_i^{\text{ASD}}$, $A_{\text{Template}}^{\text{NC}} = \frac{1}{n_2} \sum_{i=1}^{n_2} A_i^{\text{NC}}$, where $n_1$ terms the number of ASD patients, and $n_2$ denotes the number of normal subjects. Then, the connectivity difference matrix can be obtained by $A_{\text{Template}}^{\text{ASD}} - A_{\text{Template}}^{\text{NC}}$. For a comprehensive description of the time series preprocessing pipeline, please refer to Appendix C.2. Finally, the edges with the 20 largest absolute values are preserved and visualized in Fig. 3 and Fig. 11.

Overall, the differential connectivity analysis reveals widespread *hyperconnectivity* in ASD relative to NC (Supekar et al., 2013), indicating stronger functional connectivity in ASD. Specifically, enhanced connectivity is observed in the anterior cingulate cortex (ACC) and bilateral insula, which are critical regions of the salience network (Uddin et al., 2013a). Additional hyperconnectivity is detected in the frontotemporal regions, particularly in the posterior superior temporal sulcus (pSTS), which is closely associated with biological motion processing, social cognition, and speech perception (Uddin et al., 2013a), as well as in the middle temporal gyrus (MTG), motor, and visual systems. Moreover, cross-modal integration areas such as the occipito-temporal cortex (OT) and posterior middle temporal gyrus (pMTG) also exhibit increased connectivity (Hong et al., 2019). At the subsystem level, ASD patients demonstrate higher within-system connectivity in primary sensory, paralimbic, and association networks, along with stronger between-system connectivity across sensory–paralimbic, sensory–association, and paralimbic–association systems (Supekar et al., 2013). By aligning with prior reports of atypical functional connectivity in ASD, these results provide further evidence for the biological interpretability of BRep. Additional biologically interpretable analyses can be found in Appendix C.6.

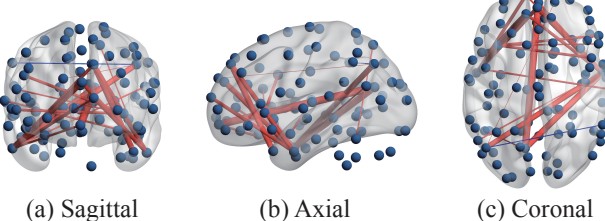

| (a) Sagittal | (b) Axial | (c) Coronal |

Figure 3: Performance variations for varying layer.

Figure 4: Performance variations for varying dimension.

**Time Series Comparison.** The experiment aims to evaluate whether different models can enhance the discriminative temporal patterns between ASD and NC. To this end, the time series are first divided into ASD and NC groups and then processed by the respective models (original, FSTA-EC, and BRep). Differential time series are computed following the procedure described in the **Case Study**, and the resulting matrices are averaged over the ROI dimension to derive group-level differential curves. Detailed processing steps are provided in Appendix C.2. Fig. 4 presents the differential time series curves obtained from the three approaches. The curve produced by FSTA-EC

Table 2: Comparison of three baseline models and their +HDM variants on ABIDE and ADHD-200 ($mean_{\pm std}$). Best models are highlighted in red.

| Model | ABIDE | | | | ADHD-200 | | | |
|---|---|---|---|---|---|---|---|---|
| | AUC ↑ | ACC ↑ | SEN ↑ | SPE ↑ | AUC ↑ | ACC ↑ | SEN ↑ | SPE ↑ |
| GCN | $59.59_{\pm 3.44}$ | $59.30_{\pm 3.38}$ | $56.67_{\pm 4.37}$ | $61.55_{\pm 5.29}$ | $67.01_{\pm 3.56}$ | $64.92_{\pm 6.32}$ | $65.09_{\pm 6.03}$ | $62.24_{\pm 5.90}$ |
| GCN+HDM | $62.06_{\pm 3.95}$ | $63.34_{\pm 8.78}$ | $64.50_{\pm 9.10}$ | $62.20_{\pm 8.40}$ | $73.24_{\pm 0.83}$ | $70.02_{\pm 2.61}$ | $75.64_{\pm 0.98}$ | $64.07_{\pm 1.68}$ |
| BrainNETTF | $77.93_{\pm 1.41}$ | $69.26_{\pm 2.26}$ | $65.92_{\pm 8.60}$ | $73.20_{\pm 6.06}$ | $79.79_{\pm 3.14}$ | $72.76_{\pm 3.17}$ | $73.64_{\pm 11.06}$ | $72.08_{\pm 5.66}$ |
| BrainNETTF+HDM | $77.61_{\pm 2.45}$ | $71.61_{\pm 3.25}$ | $68.64_{\pm 7.80}$ | $74.08_{\pm 4.84}$ | $83.04_{\pm 2.01}$ | $76.19_{\pm 2.08}$ | $75.28_{\pm 10.84}$ | $75.38_{\pm 7.67}$ |
| BQN | $79.85_{\pm 1.27}$ | $72.53_{\pm 1.41}$ | $73.26_{\pm 5.99}$ | $72.03_{\pm 6.24}$ | $83.34_{\pm 1.13}$ | $75.68_{\pm 1.95}$ | $79.73_{\pm 2.27}$ | $71.63_{\pm 4.87}$ |
| BQN+HDM | $77.59_{\pm 1.85}$ | $73.15_{\pm 1.37}$ | $73.95_{\pm 4.15}$ | $72.56_{\pm 4.68}$ | $83.44_{\pm 1.01}$ | $79.97_{\pm 3.34}$ | $76.97_{\pm 5.81}$ | $73.25_{\pm 9.84}$ |

closely resembles the raw time series, suggesting limited enhancement. In contrast, BRep yields more pronounced fluctuations and distinctive patterns, implying that it captures additional discriminative temporal dynamics beyond those present in the original signals. In summary, these findings suggest that BRep provides stronger representational capacity for distinguishing ASD from NC. Details can be found in Appendix C.3.

**Applicability Analysis.** Although Tab. 1 has demonstrated the effectiveness of the proposed BRep based on NNs (MLP and BQN), this experiment further extend the evaluation to advanced GNNs and GTs. Tab. 2 presents the results of comparing GCN and BrainNETTF with their +HDM counterparts on ABIDE and ADHD-200 dataset. It can be observed that there are consistent improvements across all metrics. In particular, GCN+HDM achieves an ACC gain of approximately $5\%$ and a SEN gain of over $10\%$ on ADHD-200. These improvements confirm that integrating HDM enhances the performance of not only *NNs* but also *GNN-* and *GT*-based models, underscoring its broader effectiveness.

**Analysis of the Mapping of High-order Dependence Measure (HDM).** Fig. 5 illustrates the impact of the dimension $D$ of the HDM on model performance. The results show that peak performance is achieved when $D = 100$. This is consistent with the theoretical statement in Section 2.2.3, since the time series dimension is also 100, which makes the parameter matrix $\mathbf{O}$ square. Such a configuration is expected to better capture correlations and thereby improve classification. In contrast, smaller dimensions tend to re-

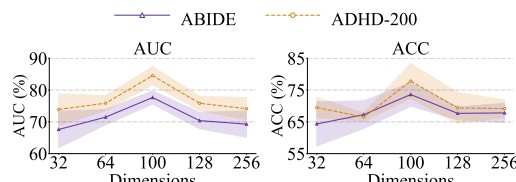

Figure 5: Impact of the dimension of the mapping of high-order dependence measure (HDM).

duce performance, likely due to underfitting, whereas excessively large dimensions (e.g., 256) may lead to a decline in performance, possibly due to overfitting. In summary, these confirm that aligning the dimension of the high-order correlation estimator with that of the timeseries is important for optimizing model capacity and generalization.

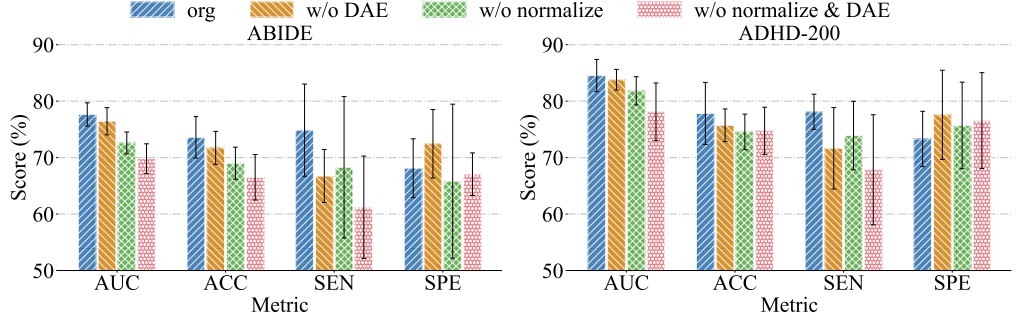

Figure 6: Ablation study on the denoising module and normalization on ABIDE and ADHD-200.

### 3.3 ADDITIONAL ANALYSIS

**Ablation Studies.** The purpose of this experiment is to evaluate the contributions of the employed denoising module and normalization strategy. To this end, three ablation variants are designed by

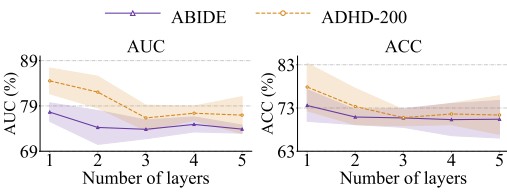
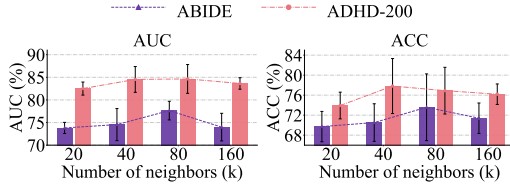

Figure 7: Impact of the number of HDM layers.

Figure 8: Impact of $K$ in TopK.

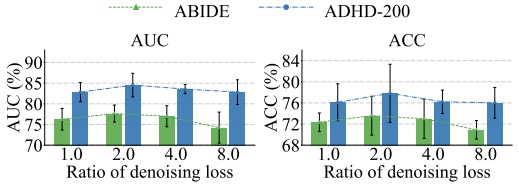
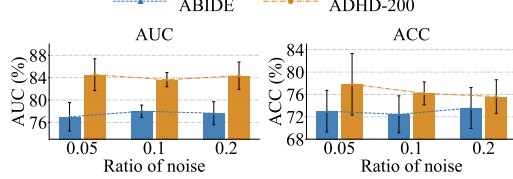

Figure 9: Impact of the denoising loss weight.

Figure 10: Impact of the noise ratio.

removing denoising, normalization, or both, and compare their performance with the original BRep on the ABIDE and ADHD-200 datasets (Fig. 6). The results show consistent performance degradation when either component is removed. In particular, eliminating both normalization and denoising leads to the largest drop, with ACC reduced by about $7\%$ and AUC by $7.84\%$ compared with the full model. Moreover, removing only one component (either denoising or normalization) still causes noticeable decreases across multiple metrics, especially in SEN and SPE. These results indicate that both components play important roles in stabilizing training and enhancing discriminative power.

**Hyperparameter Analysis.** This experiment aims to examine the impact of certain parameters on model performance. Four hyperparameter sensitivity experiments are as follows.

*(1) Number of HDM layers.* Fig. 7 shows that a single-layer HDM achieves the best performance, while deeper HDMs (2–5 layers) result in slightly lower but stable outcomes. This suggests that excessive depth does not benefit representation learning in this setting.

*(2) TopK connection.* As illustrated in Fig. 8, performance generally improves as $k$ increases from 20 to 80, with the best results observed at $k = 80$ on both datasets. This highlights the importance of appropriate graph sparsification for constructing reliable functional connectivity matrices.

*(3) Denoising loss weight.* Fig. 9 shows that the optimal performance occurs when the denoising loss weight is set to 2. This validates the contribution of the denoising component and emphasizes the need for balancing reconstruction and classification objectives: too small a weight diminishes the denoising effect, whereas too large a weight overemphasizes it.

*(4) Noise ratio.* According to Fig. 10, model performance remains robust when the noise ratio varies between 0.05 and 0.2. On ADHD-200, the best performance is observed at 0.05, while on ABIDE, results are largely stable across all noise levels. This demonstrates the robustness of the model in reconstructing time series even under moderate noise perturbations.

## 4 CONCLUSIONS

This work has introduced the Brain Representation (BRep) learning problem by rethinking the construction of brain functional networks. Rather than relying on fixed, hand-crafted correlations of BOLD time series, the study enhances linear and nonlinear correlations into high-order, parametric, and learnable forms, further combined with a TopK sparsification strategy. In this way, the brain network serves as a flexible, graph-structured representation, enabling the predictor to remain simple while maintaining an end-to-end framework. Extensive evaluations have demonstrated that the proposed BRep achieves superior predictive performance, high efficiency, and interpretability. These results indicate that transitioning from handcrafted correlations to learnable brain representations provides a promising direction for advancing network-based neurological disorder analysis. Future work may focus on incorporating temporal dynamics and heterogeneous graph structures, as well as extending BRep to multimodal neuroimaging data. Further exploration of interpretability and robustness will also be critical for establishing stronger theoretical and practical foundations.

## ETHICAL STATEMENT

Our work introduces BRep, a learnable brain network modeling framework designed to improve the diagnosis of neurological disorders while ensuring scientific rigor and respect for ethical standards. BRep is developed solely for research and healthcare-related applications, and should not be misused for purposes that could cause harm, such as unauthorized medical predictions or discriminatory practices. All experiments are conducted on publicly available datasets, with proper adherence to their corresponding usage licenses. We emphasize that any deployment of BRep must comply with ethical research practices, community guidelines, and relevant legal frameworks.

## REPRODUCIBILITY STATEMENT

To ensure reproducibility, we provide complete implementation details of BRep, including environment configuration and instructions, the network construction module, training settings, and evaluation protocols across datasets. All hyperparameters (e.g., dimension of correlation estimator, DAE loss weight, and noise ratio) are explicitly specified in the paper or appendix. The source code is publicly available at `https://anonymous.4open.science/r/BRep-demo/`.

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

## A RELATED WORKS

Resting-state fMRI functional connectivity has provided a foundation for characterizing large-scale brain networks. Building on this, graph-based deep learning methods, including Graph Neural Networks (GNNs), Graph Transformers (GTs), and Neural Networks (NNs), have been applied to brain network modeling, while effective connectivity (EC) studies offer a complementary causal perspective.

**Resting-State fMRI Functional Connectivity.** Resting-state fMRI (fMRI) has become a widely used tool for mapping large-scale brain networks and individual differences (Van Den Heuvel & Pol, 2010). Since the seminal work of Biswal et al. (1995), functional connectivity (FC) has been inferred from low-frequency blood-oxygen-level-dependent (BOLD) fluctuations (0.01–0.1 Hz) (Cordes et al., 2001). Common approaches include correlation-based analysis, independent component analysis (ICA), clustering, and graph-theoretical modeling (Beckmann et al., 2005; Achard et al., 2006). fMRI FC has provided valuable insights into cognition and has been linked to various neurological and psychiatric disorders, such as Alzheimer's disease and schizophrenia (Greicius et al., 2004; Fair et al., 2009). However, existing fMRI–based FC suffers from limited representational capacity, making it difficult to capture complex spatiotemporal dependencies among brain regions (ROIs) (Friston, 2011). **Graph Neural Networks (GNNs)** have been widely applied to FC analysis in recent years, aiming to learn representations of ROIs and connectivity patterns. Representative works include BrainGNN (Li et al., 2021), BrainGB (Cui et al., 2022), FBNetGen (Kan et al., 2022a) and A-GCL (Zhang et al., 2023). These methods typically rely on local neighborhood aggregation over FC-derived graphs, which may overlook global interactions among ROIs. **Graph Transformers (GTs)** were subsequently introduced to address this limitation. By employing global self-attention, GTs are able to capture holistic inter-regional interactions and long-range dependencies. Notable examples include BrainNETTF (Kan et al., 2022b), ContrastPool (Xu et al., 2024), ALTER (Yu et al., 2024), and BioBGT (Peng et al., 2025). However, these approaches often conflate functional connectivity matrices with ROIs features, where correlation coefficient matrices are simultaneously treated as both adjacency matrices and node features. More recently, **Neural Networks (NNs)** have been proposed to address this issue by treating the Pearson Correlation Coefficient matrix as a single input and employing non–message-passing mechanisms to model brain functional connectivity. For instance, BQN (Yang et al., 2025) introduces a simple yet effective Quadratic Neural Network specifically designed for modeling brain functional connectivity.

**Brain Effective Connectivity.** Early studies on effective connectivity (EC) relied on model-driven methods such as SEM (Eisenhauer et al., 2015), DCM (Friston et al., 2012; 2014), and GC (DSouza et al., 2017), which provide causal interpretability but depend heavily on prior assumptions. Subsequent approaches, including CTE-score (Liu et al., 2021) and VACOEC (Liu et al., 2019), sought to better capture temporal information and improve search efficiency. Recently, deep learning models like STGCMEC (Zou et al., 2022) and FSTA-EC (Xiong et al., 2025) have emerged, marking a shift from model-driven to data-driven paradigms in EC research.

## B EXPERIMENTAL SETUP

**Datasets.** The experiments utilize four benchmark fMRI datasets for brain network analysis:

- Autism Brain Imaging Data Exchange (**ABIDE**)[1]. It combines functional and structural brain imaging data from seventeen international sites to study the neural bases of autism. It comprises 1,009 subjects, including 516 individuals with Autism Spectrum Disorder (ASD) and 493 normal controls (NC) (Craddock et al., 2012).

- Attention Deficit Hyperactivity Disorder (**ADHD-200**)[2]. It is a multi-site dataset to study neural basis of Attention Deficit Hyperactivity Disorder (ADHD-200). The experiment utilizes 459 subjects, specifically, 230 developing individuals and 229 ADHD patients.

- Alzheimer's Disease Neuroimaging Initiative (**ADNI**)[3]. It is a multi-center, longitudinal research program designed to systematically collect cognitive assessments, neuroimag-

---

[1] http://preprocessed-connectomes-project.org/abide/
[2] https://fcon_1000.projects.nitrc.org/indi/adhd200/
[3] https://adni.loni.usc.edu/

ing data, and associated biomarkers to enable quantitative analysis of patterns related to Alzheimer's disease progression. The subset used in this study comprises 538 subjects, categorized into four predefined groups with sample sizes of 54, 79, 194, and 211, respectively.

- Parkinson's Progression Markers Initiative (**PPMI**). It is a large-scale, multi-center longitudinal study aimed at identifying biological markers associated with Parkinson's disease risk, onset, and progression. The dataset encompasses four clinically distinct subject groups: 15 normal controls (NC), 14 individuals with scans without evidence of dopaminergic deficit (SWEDD), 67 prodromal subjects, and 113 patients diagnosed with Parkinson's disease (PD).

Following Craddock et al. (2012), ROIs for both datasets are defined using the Craddock 200 atlas, with 200 ROIs in ABIDE and 190 ROIs in ADHD-200. For ADNI, resting-state fMRI data were preprocessed using the Data Processing Assistant for Resting-State fMRI (DPARSF) toolkit, followed by region definition based on the AAL90 atlas. For PPMI, regions of interest were delineated using the AAL116 atlas, and all preprocessing procedures were completed by Xu et al. (2023). Following Yang et al. (2025); Kan et al. (2022b), both datasets are randomly split into training, validation, and test sets with a ratio of $7/1/2$.

**Baselines.** The proposed BRep is evaluated by comparing its performance with 14 baseline models. Based on their architecture, they are grouped into the following three categories.

- Graph Neural Networks (*GNNs*), including classic GCN (Kipf, 2016) and GAT (Veličković et al., 2017), and four brain-specific GNNs, *i.e.*, BrainGNN (Li et al., 2021), BrainGB (Cui et al., 2022), FBNetGen (Kan et al., 2022a), and A-GCL (Zhang et al., 2023);
- Graph Transformers (*GTs*), including three general GTs, that is, SAN (Kreuzer et al., 2021), Graphormer (Ying et al., 2021), and GraphTrans (Wu et al., 2021), and four brain-specific GTs, *i.e.*, BrainNETTF (Kan et al., 2022b), ContrastPool (Xu et al., 2024), ALTER (Yu et al., 2024), and BioBGT (Peng et al., 2025);
- Neural Network (*NNs*), that is, MLP (Rumelhart et al., 1986) and BQN (Yang et al., 2025).

**Metrics.** The aim is to diagnose ASD on the ABIDE dataset and ADHD on the ADHD-200 dataset, both formulated as binary classification tasks. Considering the biomedical nature, the model is evaluated using four metrics: (1) Area Under the ROC Curve (AUC), quantifying the trade-off between true positive and false positive rates; (2) Accuracy (ACC), indicating the proportion of correctly classified samples; (3) Sensitivity (SEN), quantifying the model's ability to identify positive cases; and (4) Specificity (SPE), indicating the model's ability to identify negative cases.

For multi-class datasets such as ADNI and PPMI, and considering their biomedical nature, the evaluation is conducted using five metric: (1) Area Under the ROC Curve (AUC), computed using the one-vs-rest scheme and macro-averaged across classes; (2) Accuracy (ACC), measuring the overall proportion of correctly classified samples; (3) F1 score (F1), calculated as the macro-average of per-class F1 scores; (4) Sensitivity (SEN), obtained as the macro-average of per-class recall; (5) Specificity (SPE), derived from the multi-class confusion matrix and macro-averaged across classes.

**Implementation Details.** All experiments are conducted in PyTorch on a Linux machine equipped with a single GeForce RTX 3090 24GB GPU. The models are tuned under a semi-supervised learning framework, and hyperparameters are selected via a grid search strategy. All baseline models are reproduced using the hyperparameters reported in their original papers. The Adam optimizer is used with an initial learning rate of $1 \times 10^{-4}$, a target learning rate of $1 \times 10^{-5}$, and a weight decay of $1 \times 10^{-3}$. Both cross-entropy and mean squared error (MSE) losses are utilized for training. Dropout rates are selected from the set $\{0.0, 0.1, 0.2, 0.3\}$. Final results are reported as the mean and standard deviation over five random runs.

## C EXPERIMENTAL SUPPLEMENT

### C.1 CIRCLE PLOTS OF DIFFERENTIAL BRAIN CONNECTIONS

Fig. 11 (a) offers a complementary circular visualization with more detailed brain-region annotations. Specifically, it depicts the top-20 differential connections, where red and blue lines denote

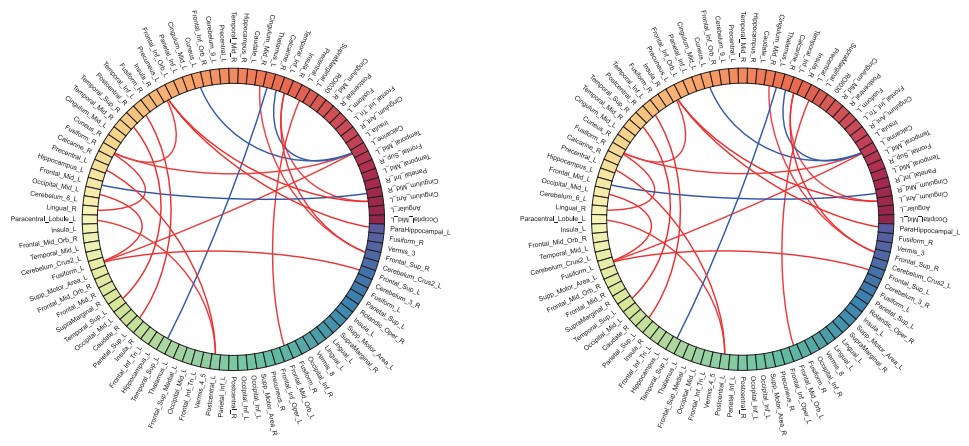

(a) Circular visualization of the **original** top-20 differential connections

(b) Circular visualization of the top-20 **permutation-based** differential connections

Figure 11: Circular visualization of the **original** and **permutation-based** top-20 differential connections between ASD and NC identified by BRep. Red and Blue lines represent positive and negative values, respectively.

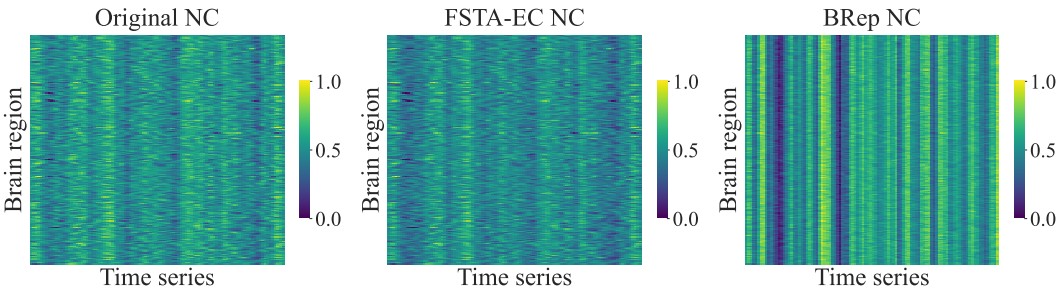

Figure 12: Time Series heatmaps of NC subjects under different processing methods (Original, FSTA-EC, and BRep)

positive and negative values, respectively. This representation offers a clearer view of how the connections are distributed across specific cortical and subcortical regions, thereby enhancing the interpretability of the results.

To ensure the statistical rigor of the interpretability analysis, we conducted 2000 edge-wise permutation testing on the BRep learned matrix $A$ of all 493 ASD and 516 NC, followed by Benjamini–Hochberg FDR correction (q<0.05), as shown in Fig. 11 (b). Only connections that remained significant after correction were retained. The resulting set of significant edges shows strong overlap with those exhibiting the largest raw ASD – NC differences, indicating that the model-identified group differences are stable, reliable, and unlikely to be driven by noise. Furthermore, the perturbed top-20 difference matrix closely matches the original top-20 difference matrix, demonstrating that the observed patterns are robust to random fluctuations.

## C.2 PROCESSING OF DIFFERENTIAL TIME SERIES

To compute the differential curves for the three models (Original, FSTA-EC, and BRep), the original time series are split into ASD and NC groups: $T_{\text{ASD}} \in \mathbb{R}^{b_1 \times n \times t}$ and $T_{\text{NC}} \in \mathbb{R}^{b_2 \times n \times t}$, where $b_1$, $b_2$ denote the numbers of ASD and NC subjects, $n$ terms the number of ROIs, and $t$ stands for the dimension of each time series. For each model, the processed time series are $T_*^{\text{ASD}} \in \mathbb{R}^{b_1 \times n \times t}$, $T_*^{\text{NC}} \in \mathbb{R}^{b_2 \times n \times t}$, where $*$ can be *org, FAST-EC, BRep*, which stand for the orgional time series, FAST-EC processed and BRep processed time series, respectively. Group-level templates are obtained by averaging across subjects: $T_{\text{ASD}}^{\text{Template} *} = \frac{1}{b_1} \sum_{i=1}^{b_1} T_*^{\text{ASD}}{}_i \in \mathbb{R}^{n \times t}, T_{\text{NC}}^{\text{Template} *} = \frac{1}{b_2} \sum_{i=1}^{b_2} T_*^{\text{ASD}}{}_i \in \mathbb{R}^{n \times t}$.

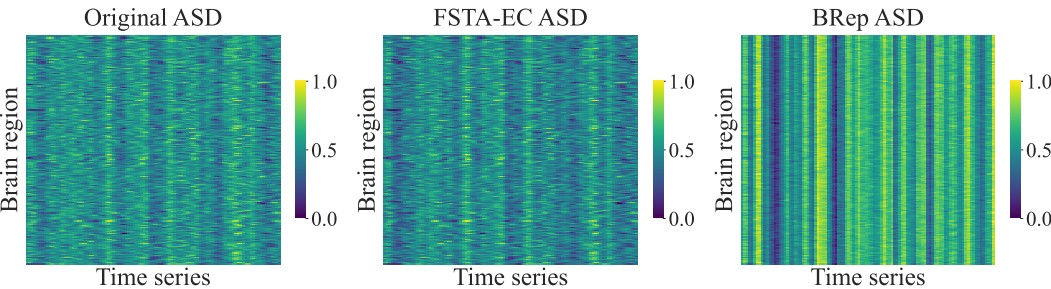

Figure 13: Time Series heatmaps of ASD subjects under different processing methods (Original, FSTA-EC, and BRep)

Table 3: Comparison of four baseline models and +HDM variants on multi-class datasets(ADNI, PPMI) ($mean_{\pm std}$). Best models are highlighted in red.

| Model | ADNI | | | | | PPMI | | | | |
|---|---|---|---|---|---|---|---|---|---|---|
| | AUC ↑ | ACC ↑ | F1 ↑ | SEN ↑ | SPE ↑ | AUC ↑ | ACC ↑ | F1 ↑ | SEN ↑ | SPE ↑ |
| GCN | $53.99_{\pm1.51}$ | $41.67_{\pm4.84}$ | $22.54_{\pm1.95}$ | $36.31_{\pm2.35}$ | $75.77_{\pm1.41}$ | $59.58_{\pm4.86}$ | $56.00_{\pm3.90}$ | $23.85_{\pm4.00}$ | $27.96_{\pm2.44}$ | $76.89_{\pm1.60}$ |
| GCN+HDM | $60.46_{\pm2.19}$ | $41.92_{\pm5.10}$ | $23.98_{\pm3.20}$ | $32.32_{\pm3.05}$ | $76.48_{\pm1.88}$ | $65.31_{\pm5.34}$ | $54.08_{\pm7.42}$ | $27.33_{\pm5.22}$ | $30.02_{\pm3.54}$ | $78.38_{\pm2.30}$ |
| BrainNETTF | $63.86_{\pm1.84}$ | $50.40_{\pm1.67}$ | $30.47_{\pm3.32}$ | $34.34_{\pm1.70}$ | $80.28_{\pm0.64}$ | $59.08_{\pm2.19}$ | $51.67_{\pm6.24}$ | $25.86_{\pm4.92}$ | $27.64_{\pm2.70}$ | $75.07_{\pm1.24}$ |
| BrainNETTF+HDM | $68.50_{\pm1.88}$ | $52.70_{\pm3.50}$ | $31.85_{\pm2.12}$ | $36.90_{\pm2.41}$ | $82.90_{\pm1.43}$ | $61.03_{\pm3.93}$ | $52.50_{\pm5.24}$ | $32.98_{\pm8.92}$ | $34.64_{\pm6.70}$ | $78.01_{\pm1.77}$ |
| BQN | $65.49_{\pm1.92}$ | $51.76_{\pm2.73}$ | $30.28_{\pm3.32}$ | $35.49_{\pm2.77}$ | $80.83_{\pm1.16}$ | $67.23_{\pm3.20}$ | $50.42_{\pm6.64}$ | $26.48_{\pm7.52}$ | $33.95_{\pm6.80}$ | $76.13_{\pm1.62}$ |
| BQN+HDM | $68.53_{\pm0.88}$ | $54.57_{\pm1.09}$ | $31.03_{\pm0.47}$ | $36.22_{\pm0.61}$ | $81.87_{\pm0.37}$ | $70.60_{\pm2.34}$ | $57.50_{\pm3.75}$ | $29.81_{\pm2.62}$ | $33.14_{\pm2.45}$ | $78.92_{\pm1.25}$ |
| MLP | $66.34_{\pm1.66}$ | $51.59_{\pm3.86}$ | $29.84_{\pm2.05}$ | $34.22_{\pm2.10}$ | $80.39_{\pm1.46}$ | $67.41_{\pm7.36}$ | $51.67_{\pm9.45}$ | $28.04_{\pm6.46}$ | $28.84_{\pm4.93}$ | $76.31_{\pm2.55}$ |
| BRep(MLP+HDM) | $69.31_{\pm1.32}$ | $55.80_{\pm2.16}$ | $32.45_{\pm1.30}$ | $37.52_{\pm1.44}$ | $83.37_{\pm0.87}$ | $72.22_{\pm4.51}$ | $59.58_{\pm5.37}$ | $28.97_{\pm1.98}$ | $31.36_{\pm1.95}$ | $79.72_{\pm1.04}$ |

Finally, group-level templates are averaged over ROIs, and their difference yields the differential time series template for each model, that is, $T_{\text{differential}}^{\text{Template *}} = \frac{1}{n} \sum_{i=1}^{n} \left( T_{\text{ASD}}^{\text{Template *}} \ominus T_{\text{NC}}^{\text{Template *}} \right)_i \in \mathbb{R}^t$.

## C.3 VISUALIZATION OF TIME SERIES PATTERNS BETWEEN ASD AND NC

Fig. 12 and 13 presents the heatmaps of NC and ASD time series under different processing methods. Compared to the Original and FSTA-EC time series, the BRep-processed time series exhibit clearer and more structured patterns, with reduced noise and enhanced regularity. This suggests that BRep is able to effectively extract informative signals while suppressing irrelevant variations, demonstrating superior denoising capability and stronger representation learning capacity.

## C.4 EXPERIMENTS ON MULTI-CLASS BRAIN DISORDER CLASSIFICATION

Given that the main body of the paper evaluates only a limited set of datasets, additional experiments on the ADNI and PPMI multi-class datasets are included in this section.

Comparison of four baseline models and +HDM variants on ADNI, PPMI are showed in Tab. 3. The results in the table show that the +HDM variant outperforms the original model across nearly all metrics and architectures. Moreover, BRep achieves consistently strong performance on both datasets.

## C.5 COMPARISON BETWEEN BREP'S LEARNED FC AND NON-END-TO-END FC PATTERNS

To compare BRep's learned FC with other non-end-to-end FC patterns, this section employs other method's learned connectivity as input to the same MLP predictor (i.e., GAT+MLP, Graphormer+MLP, and BRep+MLP), ensuring a fair and consistent evaluation. As reported in the Fig. 14, the MLP equipped with BRep's learned FC achieves the best performance, indicating that the connectivity learned by BRep is more informative than that obtained by GAT or Graphormer. Note that GAT and Graphormer still rely on hand-crafted Pearson Correlation matrix with attention, whereas BRep learns FC directly from time series in an end-to-end fashion.

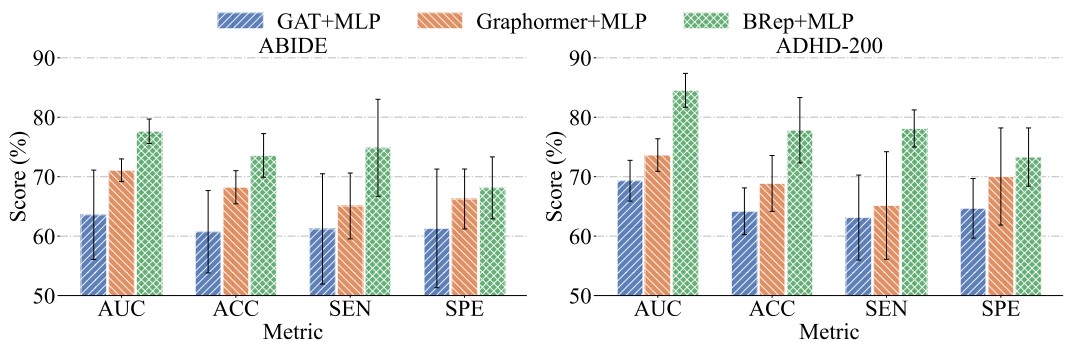

Figure 14: Comparison between BRep's FC and non-end-to-end FC patterns on ABIDE and ADHD-200.

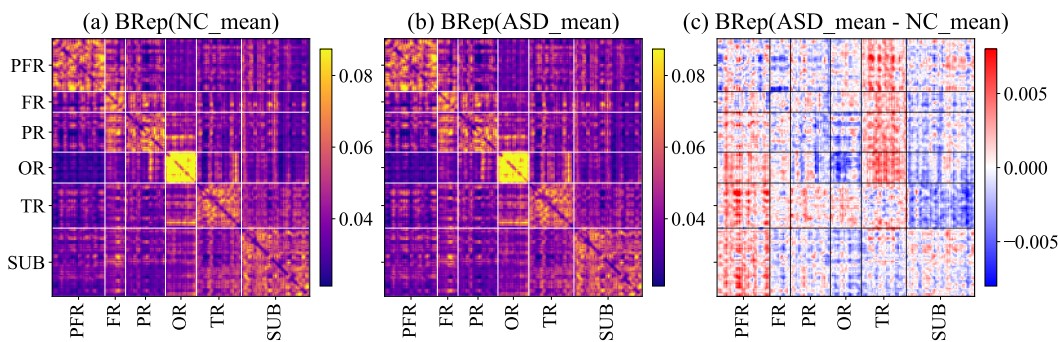

Figure 15: BRep-learned NC, ASD, and differential brain connectivity heatmaps.

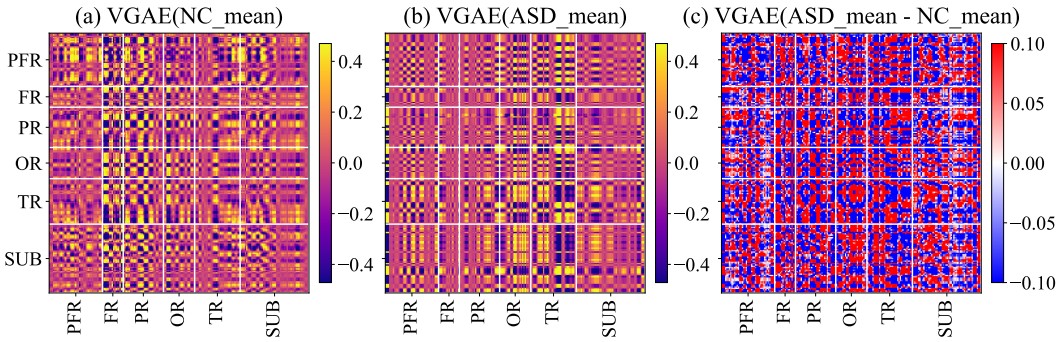

Figure 16: VGAE-learned NC, ASD, and differential brain connectivity heatmaps.

## C.6 COMPARASION OF BREP-LEARNED AND VGAE-LEARNED BRAIN CONNECTIONS

To further assess the reliability of the BRep-learned brain connectivity, this section presents the NC, ASD brain connections matrices and the differential brain connections computed from the BRep-learned matrices and VGAE-learned (Kipf & Welling, 2016) matrices after averaging the NC and ASD groups.

Fig. 15 and 16 show the BRep-learned and VGAE-learned NC, ASD brain connections matrices and ASD − NC differential brain connection on ABIDE. To clearly demonstrate the biological interpretability of BRep-learned brain connections, all ROIs are reordered into six atlas-based macro–anatomical groups – Prefrontal (PFR), Frontal (FR), Parietal (PR), Occipital (OR), Temporal (TR), and Subcortical (SUB).

The BRep-based NC and ASD templates (Fig. 15 (a) and (b)) show a recognizable modular structure: within-block connectivity (block diagonals) is stronger than between-block connectivity

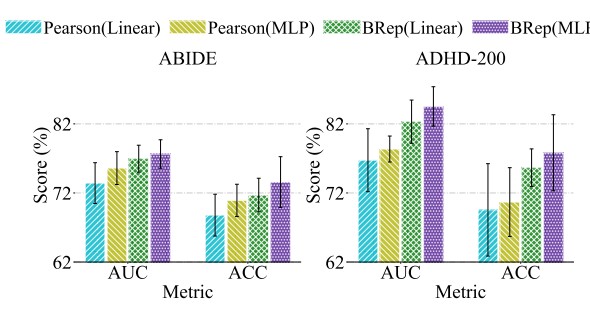

Figure 17: Comparison of Pearson and BRep with linear and MLP predictors

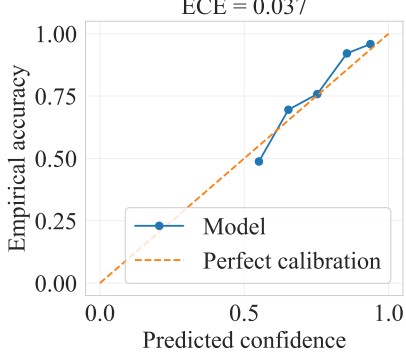

Figure 18: Reliability Diagram and ECE on ABIDE

(Power et al., 2011).The NC and ASD brain connections remain visually similar, which is consistent with ASD being a neurodevelopmental condition with subtle, distributed FC alterations rather than gross disruption of whole networks (Holiga et al., 2019). Note that the BRep-learned differential brain connection (Fig. 15 (c)) shows spatially coherent clusters of altered connectivity, mainly in TR–FR/PFR, TR–OR, TR–PR, and SUB–PFR connections, corresponding to atypical coupling between sensory, visual, social–cognitive, and subcortical–prefrontal systems that have frequently been reported in ASD studies (Supekar et al., 2013; Long et al., 2016; Uddin et al., 2013b; Woodward et al., 2017). These patterns are therefore interpretable from a neuroscience perspective rather than appearing as random noise.

In contrast, the VGAE-based maps (Fig. 16), under the same ordering, do not exhibit clear FC pattern or strong within-module cohesion; both the brain connections and the ASD – NC difference are dominated by rapid sign changes at the single-ROI level with little alignment to lobar/subcortical groups. This lack of modular and spatially coherent structure makes the VGAE-learned connectivity harder to relate to known large-scale functional networks.

Taken together, these observations suggest that BRep tends to learn FC patterns that are more structured, spatially coherent, and consistent with known large-scale functional systems, whereas the VGAE results in our experiments appear more fragmented, with ROI-level fluctuations that are difficult to interpret neurobiologically. This qualitative comparison indicates that BRep can provide more interpretable and neuroscientifically meaningful connectivity estimates than VGAE.

### C.7 PERFORMANCE OF BREP UNDER LINEAR AND NONLINEAR SETTINGS

To test the performance benefits of BRep across both linear and nonlinear settings, we evaluate four combinations: (i) conventional Pearson correlation connectivities with a linear probe, (ii) Pearson with a nonlinear MLP, (iii) BRep connectivities with a linear probe, and (iv) BRep (with a nonlinear MLP). As Fig. 17 shows, in the linear probing setting, BRep consistently provides an advantage over conventional correlation connectivities on both ABIDE and ADHD-200, while its performance does not match that of the nonlinear MLPs. This indicates that the gain of BRep comes from the learned BRep-based connectivity itself rather than merely from using a more expressive predictor.

## D RISK CALIBRATION ANALYSIS

To evaluate whether the predicted probabilities are reliable at the individual level, this study conducts a risk calibration analysis. A reliability diagram is generated using the model's predicted confidence scores, and the Expected Calibration Error (ECE) is computed to quantify calibration performance. As shown in Fig. 18, the calibration curve closely aligns with the ideal diagonal line across the full confidence range, indicating strong consistency between the predicted probabilities and the empirical positive rates. On the ABIDE, the model achieves ECE of 0.037, which demonstrates excellent calibration quality and indicates that the predicted risk scores accurately reflect each subject's true likelihood of disease.

Table 4: Paired t-test results comparing BRep with baseline models

| Metrics | Model | ABIDE | | | ADHD-200 | | |
|---------|-------|---|----|---|---|----|---|
| | | $t$ | $df$ | $p$ | $t$ | $df$ | $p$ |
| *AUC* | BRep – BQN | 2.7412 | 9 | 0.0228 | 2.6709 | 9 | 0.0256 |
| | BRep – BrainNETTF | 2.8141 | 9 | 0.0202 | 2.9019 | 9 | 0.0175 |
| *ACC* | BRep – BQN | 2.5711 | 9 | 0.0301 | 2.8046 | 9 | 0.0206 |
| | BRep – BrainNETTF | 2.7777 | 9 | 0.0215 | 2.5609 | 9 | 0.0306 |

## E  STATISTICAL SIGNIFICANCE ANALYSIS

To assess the statistical significance of the performance improvements of BRep over baseline models, paired t-tests were conducted on 10 fully matched runs ($df = 9$) under the optimal hyperparameter settings, using a significance threshold of $\alpha = 0.05$. Tab. 4 reports the resulting t-values and p-values for AUC and ACC on both the ABIDE and ADHD-200 datasets. All comparisons satisfy $p < 0.05$, indicating statistically significant performance gains. The t-values, ranging from 2.56 to 2.90, quantify the magnitude of the performance difference relative to the variability across the paired runs. Overall, the results in Tab. 4 demonstrate that BRep provides more stable and consistently stronger discriminative performance compared with BQN and BrainNETTF.

## F  PROOFS OF THEOREMS

### F.1  PROOF OF THEOREM 2.1

*Proof.* The map $S : (\mathbf{x}, \mathbf{y}) \mapsto (\mathbf{u}, \mathbf{v}) = (\mathbf{W}\mathbf{x}, \mathbf{W}\mathbf{y})$ is a linear homeomorphism of $\mathbb{R}^{2m}$ because $\mathbf{W}$ is invertible; in particular $S$ restricts to a homeomorphism $S : K \to S(K)$ and $S(K)$ is compact. The mapping

$$(\mathbf{x}, \mathbf{y}) \mapsto (\mathbf{u}, \mathbf{v}, \text{vec}(\mathbf{u}\mathbf{v}^\top))$$

is continuous (it is a composition of linear maps and polynomial multiplications); denote it by $T$. Since $S$ is injective on $K$ and $T$ includes $(\mathbf{u}, \mathbf{v})$ as components, $T$ is injective on $K$. Hence $T : K \to T(K)$ is a homeomorphism onto its compact image $T(K)$, and the inverse $T^{-1} : T(K) \to K$ is continuous.

Define $g_0 := h \circ T^{-1} : T(K) \to \mathbb{R}$. Then $g_0$ is continuous on the compact set $T(K)$. By the universal approximation theorem, there exists a network $g \in \mathcal{H}$ such that

$$\sup_{\mathbf{z} \in T(K)} |g_0(\mathbf{z}) - g(\mathbf{z})| < \varepsilon,$$

which implies for all $(\mathbf{x}, \mathbf{y}) \in K$,

$$|h(\mathbf{x}, \mathbf{y}) - g(T(\mathbf{x}, \mathbf{y}))| = |g_0(T(\mathbf{x}, \mathbf{y})) - g(T(\mathbf{x}, \mathbf{y}))| \leq \sup_{\mathbf{z} \in T(K)} |g_0(\mathbf{z}) - g(\mathbf{z})| < \varepsilon.$$

This proves the claim. □

### F.2  PROOF OF THEOREM 2.4

*Proof.* Fix some $\mathbf{y}_0$ such that $\{(\mathbf{x}, \mathbf{y}_0) : \mathbf{x} \in U\} \subset K$ with $U$ containing two points $\mathbf{x}^{(1)} \neq \mathbf{x}^{(2)}$ as in the statement. Suppose by contradiction that $\mathcal{F}_1$ were dense in $C(K)$. Consider the continuous function $h(\mathbf{x}, \mathbf{y}) := \mathbf{x}_1^2$ restricted to $K$. For any $M \in \mathbb{R}^{m \times m}$ and $\phi \in C(\mathbb{R})$ define $f(\mathbf{x}, \mathbf{y}) := \phi(\mathbf{x}^\top M \mathbf{y})$. Fixing $\mathbf{y} = \mathbf{y}_0$, the function $\mathbf{x} \mapsto f(\mathbf{x}, \mathbf{y}_0)$ depends on $\mathbf{x}$ only through the scalar linear projection $\ell(x) := \mathbf{x}^\top (M\mathbf{y}_0)$. Thus for any two distinct $\mathbf{x}^{(1)}, \mathbf{x}^{(2)}$ in $U$ satisfying $\ell(\mathbf{x}^{(1)}) = \ell(\mathbf{x}^{(2)})$ we have $f(\mathbf{x}^{(1)}, \mathbf{y}_0) = f(\mathbf{x}^{(2)}, \mathbf{y}_0)$ while $h(\mathbf{x}^{(1)}, \mathbf{y}_0) \neq h(\mathbf{x}^{(2)}, \mathbf{y}_0)$ provided we choose $\mathbf{x}^{(1)}, \mathbf{x}^{(2)}$ so that $\mathbf{x}_1^{(1)} \neq \mathbf{x}_1^{(2)}$.

Because the kernel of a nontrivial linear functional has positive codimension, for any fixed $M$ and $\mathbf{y}_0$ one can find distinct $\mathbf{x}^{(1)}, \mathbf{x}^{(2)}$ with $\ell(\mathbf{x}^{(1)}) = \ell(\mathbf{x}^{(2)})$ and yet $\mathbf{x}_1^{(1)} \neq \mathbf{x}_1^{(2)}$ (for example, take $\mathbf{x}^{(2)} = \mathbf{x}^{(1)} + \mathbf{z}$ with $\mathbf{z} \in \ker(M\mathbf{y}_0)$ but $\mathbf{z}_1 \neq 0$; such $\mathbf{z}$ exists for generic choices because $\ker(M\mathbf{y}_0)$

is either of dimension $\geq 1$ or the restriction on $U$ ensures existence). Thus for that $M$ and any $\phi$ we have

$$\sup_{\mathbf{x} \in U} |h(\mathbf{x}, \mathbf{y}_0) - f(\mathbf{x}, \mathbf{y}_0)| \geq |h(\mathbf{x}^{(1)}, \mathbf{y}_0) - f(\mathbf{x}^{(1)}, \mathbf{y}_0)| + |h(\mathbf{x}^{(2)}, \mathbf{y}_0) - f(\mathbf{x}^{(2)}, \mathbf{y}_0)|$$

giving a positive lower bound. Since this holds for each $M$ the family $\mathcal{F}_1$ cannot approximate $h$ uniformly within arbitrarily small error; hence $\mathcal{F}_1$ is not dense in $C(K)$. $\square$

### F.3 PROOF OF THEOREM 2.6

*Proof.* Direct computation yields

$$s_{ij}(\mathbf{x}, \mathbf{y}) = \mathbf{x}_i \mathbf{y}_j,$$

so $s(\mathbf{x}, \mathbf{y}) = \text{vec}(\mathbf{x}\mathbf{y}^\top)$ by definition of vectorization. The mapping $(\mathbf{x}, \mathbf{y}) \mapsto \text{vec}(\mathbf{x}\mathbf{y}^\top)$ is polynomial and hence continuous. On any domain where distinct $(\mathbf{x}, \mathbf{y})$ lead to distinct outer-products $\mathbf{x}\mathbf{y}^\top$ (for instance if $K$ does not identify distinct pairs through the same outer-product), the mapping is injective. Because $K$ is compact, the image $s(K)$ is compact and $s$ is a homeomorphism onto its image when restricted to a region of injectivity. Define $g_0 := h \circ s^{-1}$ on $s(K)$; $g_0$ is continuous on the compact set $s(K)$. The universal approximation theorem produces an MLP $g$ approximating $g_0$ uniformly on $s(K)$ to within $\varepsilon$. Pulling back yields the desired uniform approximation of $h$ on $K$ by $g \circ s$. $\square$

### USE OF LARGE LANGUAGE MODELS (LLMS)

We declare that Large Language Models (LLMs) were used exclusively as auxiliary tools to aid the writing of this paper, primarily for polishing, grammar checking, and improving readability. LLMs were not involved in conceptual design, theoretical formulation, experimental implementation, or result analysis. All research contributions, including methodology, experiments, and conclusions, were conceived, conducted, and validated entirely by the authors.

