# OpenReview forum: "BRep: Graph-structured Brain Representation Learning via Parametric High-order Dependence Measures"
_ICLR.cc/2026/Conference — Submitted to ICLR 2026_

### Official Review · Reviewer_LF1G · 2025-10-26

**Soundness:** 3
**Presentation:** 3
**Contribution:** 2
**Rating:** 2
**Confidence:** 2

**Summary:**

This paper proposes learning functional brain connectivity from fMRI time series via end-to-end backpropagation, rather than using fixed correlation measures. The model learns a parameter matrix O that defines pairwise connectivity as (x_i O)(x_j O)^T, from which an adjacency matrix is extracted via top-k sparsification. This learned brain network is then flattened and fed into an MLP for disorder classification. The authors argue this mirrors the shift from hand-crafted features to representation learning, enabling simpler predictors. Experiments on ABIDE and ADHD-200 datasets show competitive performance, and the learned connectivity patterns align with established neuroscience findings about autism.

**Strengths:**

- Framing  brain network construction as a representation learning problem aligns well with the spirit of deep learning.
- The paper is overall well written and the problem statement is well defined.
- The unification of linear and non-linear correlations through inner product in latent spaces is clever (section 2).
- The experimental section is thorough, including 14 baselines across GNNs, GTs, and NNs, with proper ablation studies and interpretability analysis.
- The differential connectivity analysis (Section 3.2) showing hyperconnectivity in ASD aligns with existing neuroscience literature, indicating promising interpretability and overall validating the approach.
- The approach enables simple downstream predictors (MLP) which might we valuable for high-throughput applications.

**Weaknesses:**

- Authors claim to learn high-order correlations, but what makes the interactions learned by the approach high-order? As far as I understand, what is learned is r_ij = (x_i O)(x_j O)^T, which is still a pairwise interaction between regions i and j. I suppose if there were multiple such layers, high-order interactions (that is, interactions between >2 regions) can be captured, but with a single layer that just doesn't happen, which makes me confused about the framing of the paper.
- Authors claim to achieve superior performance (abstract), but on ABIDE dataset, there are better methods, and on ADHD-200 dataset, the difference is within error bars.
- In my opinion, the contribution lacks noveltly sufficient for an ICLR contribution. The approach of learning adjacency matrices end-to-end and connecting it to functional connectivity is well-established [1,2]. Moreover, several included baselines (GAT, Graphormer, FBNetGen) already learn connectivity through attention or generation mechanisms. While these are compared numerically (Table 1), the paper does not acknowledge their learnable nature or explain how the proposed explicit parameterization via transformation matrix O differs from or improves upon these implicit learning mechanisms.
- I personally believe the work would be a much stronger contribution if it focused on the learned patterns and a thorough comparison to other methods in the aspect (e.g. [1,2], or graph structure learning approaches such as classical graph VAE from Kipf et al.). It feels to me that the contribution is more suitable for a medical journal / conference where the interpretability results might be interesting to the practitioners.

[1] Zhdanov, Maksim et al. “Investigating Brain Connectivity with Graph Neural Networks and GNNExplainer.” 2022 26th International Conference on Pattern Recognition (ICPR) (2022): 5155-5161.
[2] Kan, Xuan et al. “Brain Network Transformer.” ArXiv abs/2210.06681 (2022): n. pag.

**Questions:**

- Would it be possible to compare learned FC patterns against other frameworks with learnable connectivity?
- What makes method learn high-order interactions? Are not those just pair-wise that the method encodes?

---

> ### Author Response · Authors · 2025-11-21
> **Rebuttal part 1**
>
> > Q1&Q6. Authors claim to learn high-order correlations, but what makes the interactions learned by the approach high-order? What we call “high-order” is the dependence between the two time series x_i and x_j behind each edge weight, not the graph connectivity itself.
>
> I’m deeply sorry that the concept of high-order causes your confusion. In this paper, the high-order refers to **the methods to estimate the correlation** between a pair of brain regions, denoted as $r_{ij}$. By denoting $v_i$ and $v_j$ as the random variables associated with two brain regions, their samples are collected as $x_{i1}, x_{i2}, \cdots, x_{iM}$ and $x_{j1}, x_{j2}, \cdots, x_{jM}$. The methods for estimating the correlation $r_{ij}$ can be divided into three categories: the linear (first-order) method, the nonlinear (second-order) method, and the high-order method, **according to the number of samples employed for comparison**. Please refer to Figures 2(b)-2(c)  for a visual and intuitive explanation.
> linear (first-order) correlation: $r_{ij} = \sum_{t} x_{it} \cdot x_{jt}$ estimates the correlation by comparing **each sample** of one random variable against the corresponding one of the other random variable;
> nonlinear (second-order) correlation: $r_{ij} = \sum_{t,s} |x_{it}-x_{is}| \cdot |x_{jt}-x_{js}|$ estimates the correlation by comparing **each pair of samples** of one random variable against the corresponding pair of samples of the other random variable;
> High-order correlation: $r_{ij} = \sum_{t_1,t_2, \cdots, t_M} f\big( \left(x_{it_1}, x_{it_2}, \cdots, x_{it_M} \right), \left(x_{jt_1}, x_{jt_2}, \cdots, x_{jt_M} \right) \big)$ estimates the correlation by comparing **each $M$-tuple of samples** of one random variable against the corresponding $M$-tuple of samples of the other random variable;
>
> In summary, we estimate **correlation between a pair of random variables** by using **a high-order $M$-tuple of samples**. Therefore, we interchangeably use high-order correlation and high-order dependence measures in the original paper. To avoid ambiguity, we may remove the "high-order” before "correlation” and only employ "high-order dependence measures”.
>
>
>
>
> > Q2. Authors claim to achieve superior performance (abstract), but on ABIDE dataset, there are better methods, and on ADHD-200 dataset, the difference is within error bars.
>
> R2. We would like to clarify how our results relate to the performance claim in more detail. As shown in Table 1, BRep with a simple MLP predictor achieves the best performance in 5 out of 9 evaluation settings and is competitive (often within the error bars) in the others. When the predictor is replaced by a stronger one (e.g., BQN, which is the second-best model in Table 1), BRep+BQN still brings additional gains, as reported in Table 2, indicating that the improvement comes from BRep’s learned connectivity structure. Moreover, Table 2 shows that attaching BRep to different predictors consistently improves their performance, suggesting that further gains are possible when combined with even more powerful predictors in future work.

---

> ### Author Response · Authors · 2025-11-21
> **Rebuttal part 2**
>
> > Q3. The approach of learning adjacency matrices end-to-end and connecting it to functional connectivity is well-established. Baselines like GAT, Graphormer, FBNetGen already learn connectivity through attention or generation mechanisms.
>
> R3. **While the mentioned works do include learnable connectivity components, they differ from our setting in where the graph comes from and how it is learned**. In BQN, GAT, and Graphormer, the graph is built from a precomputed Pearson-correlation FC matrix, and the subsequent attention/message-passing operates on this fixed hand-crafted graph rather than learning FC directly from raw time series. In [1], the adjacency matrix is fixed by the physical layout of EEG electrodes, so the connectivity structure itself is not learned. FBNetGen does generate an adjacency matrix from time series, but its graph-generation module is not trained in the same fully end-to-end “time series → learned connectivity → prediction” manner as in our framework.
>
> **By contrast, BRep takes time series as input and explicitly parameterizes a transformation matrix $\mathbf{O}$ from which functional connectivity is constructed and optimized jointly with the prediction objective**. This enables end-to-end learning of FC directly from time series and allows the learned structure to be plugged into different predictors (as shown in Tables 1–2), which we consider a key conceptual and methodological advance over existing learnable-connectivity approaches.
>
>
>
> > Q4. I personally believe the work would be a much stronger contribution if it focused on the learned patterns and a thorough comparison to other methods in the aspect (e.g. BQN, [1], or GSL approaches such as VGAE[2]).
>
> R4. According to your suggestion, we add the comparsion between BRep and VGAE[2], and visualization as the form of a difference brain connection matrices. As shown in Fig. 15 and Fig. 16,  the BRep-learned differential brain connection exhibits fine-grained and spatially coherent patterns. Several rows and columns show localized deviations, reflecting abnormal connectivity associated with specific ROIs implicated in ASD. In contrast, the VGAE-learned differential brain connection displays pronounced grid-like and checkerboard artifacts, which are unlikely to correspond to any meaningful neurophysiological structure. The comparison highlights that BRep produces more stable, detailed, and neurobiologically interpretable differential connectivity patterns than VGAE.
>
>
>
> > Q5. Would it be possible to compare learned FC patterns against other frameworks with learnable connectivity?
>
> R5. Following your comment, we have conducted additional experiments to indirectly compare the learned FC patterns: for each method, we used its learned connectivity together with the same MLP predictor (GAT+MLP, Graphormer+MLP, BRep+MLP).
>
> As reported in the above Fig. 14, the MLP equipped with BRep’s learned FC achieves the best performance, indicating that the connectivity learned by BRep is more informative than that obtained by GAT or Graphormer. Note that GAT and Graphormer still rely on precomputed Pearson-correlation graphs with attention on top, whereas BRep learns FC directly from time series in an end-to-end fashion.
>
> [1] Zhdanov, M., Steinmann, S., & Hoffmann, N. (2022, August). Investigating brain connectivity with graph neural networks and GNNExplainer. In *2022 26th International Conference on Pattern Recognition (ICPR)* (pp. 5155-5161). IEEE.
>
> [2] Kipf, T. N., & Welling, M. (2016). Variational graph auto-encoders. *arXiv preprint arXiv:1611.07308*.

---

> > ### Comment · Reviewer_LF1G · 2025-11-21
> > **Response of Reviewer LF1G**
> >
> > Dear authors, I am grateful for your response and I have a couple of question to clarify things for me.
> >
> > 1) I find the definition of high-order that you provide in your response confusing. It is established in the graph DL community to refer to high-order interactions as interactions between >2 nodes *at the same time*. For example, a 3-dim tensor r_ijk would be a high-order interaction. But this is not what happens in your work, in spatial/graph sense, you do not capture high-order interactions. I might simply be unfamiliar with the intricacies and conventions of neuroscience field. To sort things out, can you please share other sources that follow the definition you stick to (that is, the number of time samples used together in the correlation function, if I understand correctly)? Alternatively, I would remove "high-order" altogether to avoid confusion, to be honest.
> >
> > 2) Thank you for the clarification. However, I remain unconvinced by the performance claims based on Table 1.Given the high standard deviations reported (often $\pm 5%$), the differences between methods may not be statistically significant. Could you please provide statistical significance tests (e.g., paired t-tests or Wilcoxon signed-rank tests) comparing BRep against the top 2-3 performing baselines for each metric? This would substantiate whether the differences are meaningful or within statistical noise.
> >
> > 3) I am not an expert in functional connectivity, so both Fig.15 and Fig.16 look like noise to me. Also I believe there might be some bias - regular patterns from BRep are said to be spatially-coherent, and regular patterns from VGAE are said to be checkerboard artifacts. Can you please provide some interpretation of those results from the neuroscience point of view? Are there some patterns that would indicate the connectivity is meaningful (e.g. known biomarkers)?
> >
> > 4) in R5, I think you might have misunderstood my question. I was asking the actual connectivity structures and their interpretations (essentially what you did in R4). I would, however, appreciate your analysis of learned patters, see previous question.
> >
> > I choose to retain my score for now, slightly leaning towards 4, but a thorough analysis of learned FC patterns of BRep might incline the score towards weak accept.

---

> ### Author Response · Authors · 2025-11-24
> **Rebuttal part 1**
>
> > Q1. Confusion about "high-order".
>
> R1. Thanks for your kind response. We acknowledge that the concept of high-order may indeed give rise to misunderstandings, since this work lies at the intersection of graph learning and statistics. While the high-order interaction refers to interactions between >2 nodes in graph learning, this high-order means employing high-order U-statistic kernels, i.e, the size of a tuple of samples larger than 2, in mathematical statistics. The theoretical foundations of U-statistics and order-m kernels are introduced in [1-3]. According to your suggestion, we will add these sources following the definition in the paper. The reference has been added in orange text at the bottom of page **4** in the revised manuscript.
>
>
>
> > Q2.  Could you please provide statistical significance tests (e.g., paired t-tests or Wilcoxon signed-rank tests) comparing BRep against the top 2-3 performing baselines for each metric?
>
> R2. To assess the statistical significance of the performance improvements of BRep over baseline models, paired t-tests were conducted on 10 fully matched runs (df=9) under the optimal hyperparameter settings, using a significance threshold of $\alpha = 0.05$. Tab. 4 (page **22**) reports the resulting t-values and p-values for AUC and ACC on both the ABIDE and ADHD-200 datasets. All comparisons satisfy $p<0.05$, indicating statistically significant performance gains.
>
> The t-values, ranging from 2.56 to 2.90, quantify the magnitude of the performance difference relative to the variability across the paired runs. Overall, the results in Tab. 4 demonstrate that BRep provides more stable and consistently stronger discriminative performance compared with BQN and BrainNETTF.
>
> Tab. 4 in revised manuscript is shown below.
>
> **ABIDE**
>
> | Metrics |             Model             |  $t$   | $df$ |  $p$   |
> | :-----: | :---------------------------: | :----: | :--: | :----: |
> |  *AUC*  |    BRep $\textendash$ BQN     | 2.7412 |  9   | 0.0228 |
> |  *AUC*  | BRep $\textendash$ BrainNETTF | 2.8141 |  9   | 0.0202 |
> |  *ACC*  |    BRep $\textendash$ BQN     | 2.5711 |  9   | 0.0301 |
> |  *ACC*  | BRep $\textendash$ BrainNETTF | 2.7777 |  9   | 0.0215 |
>
>
>
> **ADHD-200**
>
> | Metrics |             Model             |  $t$   | $df$ |  $p$   |
> | :-----: | :---------------------------: | :----: | :--: | :----: |
> |  *AUC*  |    BRep $\textendash$ BQN     | 2.6709 |  9   | 0.0256 |
> |  *AUC*  | BRep $\textendash$ BrainNETTF | 2.9019 |  9   | 0.0175 |
> |  *ACC*  |    BRep $\textendash$ BQN     | 2.8046 |  9   | 0.0206 |
> |  *ACC*  | BRep $\textendash$ BrainNETTF | 2.5609 |  9   | 0.0306 |

---

> ### Author Response · Authors · 2025-11-24
> **Rebuttal part 2**
>
> > Q3& Q4. Can you please provide some interpretation of those results from the neuroscience point of view? Are there some patterns that would indicate the connectivity is meaningful (e.g. known biomarkers)?
>
> R3. We thank the reviewer for this comment and for explicitly asking for a neuroscience-oriented interpretation. We agree that, without additional context, the original heatmaps can look like noise. In the revised manuscript, the connectivity heatmaps (for NC and ASD templates and ASD–NC difference) of BRep and VGAE are reported in Figs. 15 and 16 (page **20**), **where all ROIs are reordered into six atlas-based macro–anatomical groups—Prefrontal (PFR), Frontal (FR), Parietal (PR), Occipital (OR), Temporal (TR), and Subcortical (SUB)**. The analyses are as follows.
>
> **The BRep-based NC and ASD templates (Fig. 15(a,b)) show a recognizable modular structure: within-block connectivity (block diagonals) is stronger than between-block connectivity** [4]. The NC and ASD templates remain visually similar, which is consistent with ASD being a neurodevelopmental condition with subtle, distributed FC alterations rather than gross disruption of whole networks [5]. Note that the BRep difference map (Fig. 15(c)) shows spatially coherent clusters of altered connectivity, mainly in TR–FR/PFR, TR–OR, TR–PR, and SUB–PFR connections, corresponding to atypical coupling between sensory, visual, social–cognitive, and subcortical–prefrontal systems that have frequently been reported in ASD studies [6-9]. These patterns are therefore interpretable from a neuroscience perspective rather than appearing as random noise.
>
> **In contrast, the VGAE-based maps (Fig. 16), under the same ordering, do not exhibit clear FC pattern or strong within-module cohesion**; both the templates and the ASD–NC difference are dominated by rapid sign changes at the single-ROI level with little alignment to lobar/subcortical groups. This lack of modular and spatially coherent structure makes the VGAE-learned connectivity harder to relate to known large-scale functional networks.
>
> Taken together, these observations suggest that BRep tends to learn FC patterns that are more structured, spatially coherent, and consistent with known large-scale functional systems, whereas the VGAE results in our experiments appear more fragmented, with ROI-level fluctuations that are difficult to interpret neurobiologically. **This qualitative comparison indicates that BRep can provide more interpretable and neuroscientifically meaningful connectivity estimates than VGAE**, and we hope it helps to alleviate the your concerns about the interpretability.
>
>
>
> [1] Hoeffding, W. (1948). A class of statistics with asymptotically normal distribution. Annals of Mathematical Statistics, 19(3), 293–325.
>
> [2] Lee, A. J. (1990). U-Statistics: Theory and Practice. New York: Marcel Dekker.
>
> [3] Sejdinovic, D., Sriperumbudur, B., Gretton, A., & Fukumizu, K. (2013). Equivalence of distance and RKHS embeddings. Annals of Statistics, 41(5), 2263–2291.
>
> [4] Power, J. D., Cohen, A. L., Nelson, S. M., Wig, G. S., Barnes, K. A., Church, J. A., Vogel, A. C., Laumann, T. O., Miezin, F. M., & Schlaggar, B. L. (2011). Functional network organization of the human brain. *Neuron*, *72*(4), 665-678.
>
> [5] Holiga, Š., Hipp, J. F., Chatham, C. H., Garces, P., Spooren, W., D’Ardhuy, X. L., Bertolino, A., Bouquet, C., Buitelaar, J. K., & Bours, C. (2019). Patients with autism spectrum disorders display reproducible functional connectivity alterations. *Science Translational Medicine*, *11*(481), eaat9223.
>
> [6] Supekar, K., Uddin, L. Q., Khouzam, A., Phillips, J., Gaillard, W. D., Kenworthy, L. E., Yerys, B. E., Vaidya, C. J., & Menon, V. (2013). Brain hyperconnectivity in children with autism and its links to social deficits. *Cell reports*, *5*(3), 738-747.
>
> [7] Long, Z., Duan, X., Mantini, D., & Chen, H. (2016). Alteration of functional connectivity in autism spectrum disorder: effect of age and anatomical distance. *Scientific reports*, *6*(1), 26527.
>
> [8] Uddin, L. Q., Supekar, K., & Menon, V. (2013). Reconceptualizing functional brain connectivity in autism from a developmental perspective. *Frontiers in human neuroscience*, *7*, 458.
>
> [9] Woodward, N. D., Giraldo-Chica, M., Rogers, B., & Cascio, C. J. (2017). Thalamocortical dysconnectivity in autism spectrum disorder: An analysis of the Autism Brain Imaging Data Exchange. *Biological Psychiatry: Cognitive Neuroscience and Neuroimaging*, *2*(1), 76-84.
>
>
>
> ---
>
>
>
> We sincerely thank you again for the thoughtful follow-up comments. In the revision, we (i) clarified the use of the concept of high-order and added supporting references, (ii) added statistical significance tests for the experiment comparisons, and (iii) expanded the neuroscientific interpretation of the learned FC patterns. All corresponding changes in the manuscript are highlighted in orange for easy checking.

---

> > ### Comment · Reviewer_LF1G · 2025-11-27
> > **Response by Reviewer LF1G**
> >
> > I appreciate the authors' thorough responses and revisions. The added statistical tests, neuroscience interpretation with proper references, and clarification of the 'high-order' terminology have substantially improved the paper. Overall, the paper makes a solid contribution and the revisions have addressed my major concerns. I am raising my score to 6 (weak accept).

---

> > > ### Author Response · Authors · 2025-11-27
> > >
> > > We sincerely thank you for the insightful comments, which have clearly helped us improve the manuscript. Our communication with you has also sharpened our understanding of how to present BRep to both the graph learning and neuroscience communities. We hope that the revised version not only addresses your concerns but also gives you greater confidence in the soundness and reliability of our study.

---

### Official Review · Reviewer_aYLf · 2025-10-27

**Soundness:** 3
**Presentation:** 3
**Contribution:** 3
**Rating:** 6
**Confidence:** 4

**Summary:**

In order to solve the problem of information loss caused by the dependence on fixed manual features (such as Pearson correlation coefficient) in the construction of brain functional networks, this paper proposes a graph-structured brain representation learning framework BRep. The core idea is to regard the brain network as a learnable graph structure representation, build a high-order, parameterized and learnable dependency metric (HDM module) by unifying and expanding linear/nonlinear correlation coefficients, and realize end-to-end training by combining TopK sparseness. On two benchmark data sets, ABIDE (autism diagnosis) and ADHD-200 (ADHD diagnosis), BRep is superior to most GNN, Graph Transformer and traditional neural network baselines in AUC and ACC, and it is interpretable (biological rationality is verified by differential brain network visualization) and universal (HDM module can improve the performance of other models). The paper also verifies the effectiveness of the key components of the framework through ablation experiments and parametric analysis, and the code is open to ensure reproducibility.

**Strengths:**

1. The linear (Pearson) and nonlinear (dCor, HSIC) correlation coefficients are unified, and the parameterized matrix approximates high-order dependence, which not only retains the modeling ability of complex brain connections, but also controls the computational complexity through dimension matching, giving consideration to performance and efficiency.
2. The end-to-end framework simplifies the downstream predictor (only MLP) and reduces the deployment difficulty.
3. The visualization results of differential brain network are consistent with the existing research conclusions related to autism (such as highlighting network hyperconnection), which is biologically interpretable and meets the needs of medical scenes.
4. HDM module can be flexibly integrated into existing models such as GCN and BrainNETTF, and its performance can be improved. It provides a general feature enhancement tool for brain network analysis, not limited to a single framework.

**Weaknesses:**

1. Only based on fMRI data verification, not extended to structural imaging (sMRI, DTI), molecular imaging (PET) and other multimodal data; Moreover, it is only aimed at the binary classification task of two kinds of neurological diseases, and lacks verification in multi-classification and regression tasks (such as disease severity assessment).
2. Although it is mentioned that HDM reduces the complexity through parameter approximation, it does not directly compare the speed/memory cost with the existing efficient high-order dependency modeling methods (such as lightweight Tensor-HSIC), and it is difficult to determine its applicability in large-scale brain networks (such as ROI > 500).
3.Differential brain network analysis only focuses on Top20 differential connections, and lacks quantitative statistical verification (such as replacement test) on why these connections are related to diseases.
4.The key hyperparameters, such as the optimal K value of TopK sparseness (80) and the module dimension of HDM (100), only verify the performance optimality through experiments, and do not explain the rationality of selection in combination with the biological characteristics of brain anatomical structure or functional partition, which reduces the domain adaptability of the method.

**Questions:**

Please refer to weaknesses.

---

> ### Author Response · Authors · 2025-11-21
> **Rebutal**
>
> > Q1. Only based on fMRI data verification, not extended to structural imaging, molecular imaging and other multimodal data; And it is only aimed at the binary classification task, and lacks verification in multi-classification and regression tasks.
>
> R1. Following the reviewer’s suggestion, we have added experiments on two additional multi-class neuroimaging datasets, ADNI and PPMI. From Tab. 3, it can be observed that adding HDM consistently improves the performance of all baseline models. Moreover, BRep (MLP+HDM) achieves the best or near-best results across almost all evaluation metrics on both datasets.
>
>
>
> > Q2. No compare the speed/memory cost with the existing efficient high-order dependency modeling methods, and it is difficult to determine its applicability in large-scale brain networks.
>
> R2. The time complexity of the proposed method is $O(N^2D+ND^2)$, and the space complexity is $O(N^2+ND)$,where $N$ denotes the number of ROIs, $D$ denotes time series length. The runtimes of the model are summarized in the table below. All results were obtained under the same experimental setting, corresponding to 200 training epochs. It is evident that BRep achieves faster runtime than the lightweight Tensor-HSIC — Tensor-Sketch HSIC[1,2] across all three datasets.
>
> | dataset  | BRep  | Tensor-Sketch HSIC |
> | :------: | :---: | :----------------: |
> |  ABIDE   | 37.27 |       41.05        |
> | ADHD-200 | 18.89 |       23.36        |
> |   ADNI   | 8.95  |       10.87        |
>
> [1] Pham, N., & Pagh, R. (2013, August). Fast and scalable polynomial kernels via explicit feature maps. In *Proceedings of the 19th ACM SIGKDD international conference on Knowledge discovery and data mining* (pp. 239-247).
>
> [2] Gretton, A., Fukumizu, K., Teo, C., Song, L., Schölkopf, B., & Smola, A. (2007). A kernel statistical test of independence. *Advances in neural information processing systems*, *20*.
>
> > Q3. Differential brain network analysis only focuses on Top20 differential connections, and lacks quantitative statistical verification (such as replacement test) on why these connections are related to diseases.
>
> R3. To ensure the statistical rigor of the interpretability analysis, we have conducted edge-wise permutation testing (2,000 permutations) on the 200×200 functional connectivity matrices of ASD (n=493) and NC (n=516), followed by Benjamini–Hochberg FDR correction (q < 0.05), as shown in Fig. 11(b). Only connections that remained significant after correction were retained. The results show that these significant edges are highly consistent with those exhibiting the largest raw ASD–NC differences, indicating that the model-identified group differences are stable, reliable, and not driven by noise. Furthermore, the perturbed Top20 difference matrix closely matches the original Top20 difference matrix, demonstrating that the observed patterns are robust to random fluctuations.
>
>
>
> > Q4. The key hyperparameters do not explain the rationality of selection in combination with the biological characteristics of brain anatomical structure or functional partition, which reduces the domain adaptability of the method.
>
> R4. In practice, the optimal sparsity level varies substantially across different atlases, diseases, and study settings. To our knowledge, neuroimaging literature does not provide a universally preferred sparsity for functional connectivity, and thus the choice of *K* is guided by empirical stability rather than fixed biological constraints. The dimensionality of the HDM module is tied to the length of the time series, and its rationale has been explained in Section 2.2.3.

---

### Official Review · Reviewer_iGCz · 2025-10-31

**Soundness:** 1
**Presentation:** 2
**Contribution:** 1
**Rating:** 2
**Confidence:** 5

**Summary:**

This paper proposes BRep, a framework for graph-structured brain representation learning, where the functional brain network is no longer constructed using fixed, hand-crafted correlation measures (e.g., Pearson, dCor), but instead learned end-to-end as a parametric, high-order dependence measure. The core idea is to treat the brain connectivity matrix as a learnable representation parameterized via a trainable matrix O that maps BOLD time series into a latent space where inner products define edge weights. This is combined with TopK sparsification and a simple MLP predictor, forming an end-to-end pipeline for neurological disorder classification. The authors evaluate BRep on ABIDE and ADHD-200 datasets, reporting competitive or superior performance over a wide range of GNNs, Graph Transformers, and neural baselines, along with interpretability analyses aligning with known neurobiological findings.

**Strengths:**

1. By replacing complex GNN/GT architectures with a learnable connectivity estimator + simple MLP, the method achieves SOTA performance with reduced architectural complexity.
2. The analogy to the pre-deep-learning era of “hand-crafted features + learnable predictors” is compelling and well-articulated.

**Weaknesses:**

1. The core technical contribution a ``parametric high-order dependence measure'' is mathematically equivalent to a learnable inner product in a linearly transformed space: $ r_{ij} = (\mathbf{x}_i \mathbf{O})(\mathbf{x}_j \mathbf{O})^\top$
where $\mathbf{O} \in \mathbb{R}^{D \times D}$ is a trainable matrix. This formulation is not high-order in any statistical sense (e.g., it does not involve third- or higher-order moments, cumulants, or tensor interactions). Instead, it is a classic instance of metric learning or representation learning via linear projection, widely used in contrastive learning, Mahalanobis distance learning, and even early deep learning architectures.

2. The paper conflates nonlinear correlation (e.g., dCor, HSIC) with low-order and rebrands a bilinear form as high-order, which is technically inaccurate and misleading. True high-order dependence measures, such as those based on U-statistics over $M$-tuples ($M \geq 3$) are explicitly mentioned but then abandoned due to computational cost. The proposed approximation via $\mathbf{O}\mathbf{O}^\top$ sidesteps the actual challenge and reduces the method to a simple, well-known paradigm.

3. The paper emphasizes  interpretability  and clinical relevance, yet provides no evidence of practical clinical value. The differential connectivity analysis (Fig. 3, Fig. 11) merely reproduces known ASD biomarkers (e.g., hyperconnectivity in the salience network, pSTS), which have been reported. This is post-hoc validation, not discovery.

4. No analysis of motion artifacts, scanner differences, or missing ROIs common in real clinical fMRI.

5. ABIDE contains 17 heterogeneous sites, yet the paper uses random splits without site-stratification. Performance may collapse under realistic multi-center conditions.

6. The model reports population-level ACC/AUC, but clinicians need calibrated, individualized risk scores with uncertainty quantification absent here.

7. The term ``graph-structured representation learning'' is presented as novel, but similar ideas appear in BQN (Yang et al., ICML 2025) and FBNetGen (Kan et al., MIDL 2022a).

**Questions:**

See above

---

> ### Author Response · Authors · 2025-11-21
> **Rebuttal part 1**
>
> > Q1. The core technical contribution a ''parametric high-order dependence measure'' is mathematically equivalent to a learnable inner product in a linearly transformed space: $r_ij = (x_i O)(x_j O)^T$where is a trainable matrix. This formulation is not high-order in any statistical sense (e.g., it does not involve third- or higher-order moments, cumulants, or tensor interactions).
>
> R1. There must be a misunderstanding. $r_{ij} = (x_i O)(x_j O)^T$ is **NOT** the metric learning, but a universal approximation to the high-order U-statistics. Your misunderstanding can be attributed to the misinterpretation of the role of the trainable matrix $O$.
>
> Firstly, $r_{ij} = (x_i O)(x_j O)^T$ is **NOT** the metric learning. The $x_i$ is the collection/set of samples $x_{it}$ of the random variable $v_i$ associated with brain region $I$ instead of the vector-form representation of brain region $I$. They are very different. A set has no inherent order among its elements, and the computation of correlation, no matter linear or nonlinear, implicitly employs this characteristic. On the contrary, the components of a vector have an inherent ordering. Thus, the trainable matrix $O$ operates on the collection of samples $x_{i1}, x_{i2}, \cdots, x_{iM}$ to combine them, instead of operating on the feature dimension of samples. Take one column of $O$ as an example, $x_i o_p = \sum_{t=1}^M o_{pt} x_{it}$ is the combination of $x_{i1}, x_{i2}, \cdots, x_{iM}$ with weights $o_{p1}, o_{p2}, \cdots, o_{pM}$, and $x_i o_p$ possesses the same dimension as samples $x_{it}$’s. Therefore, the proposed $r_{ij} = (x_i O)(x_j O)^T$ is **NOT** an instance of metric learning via linear projection.
>
> Secondly, $r_{ij} = (x_i O)(x_j O)^T$ is a universal approximation to the high-order U-statistics.  Intuitively, the characteristic of $x_i O$ on combining samples, i.e., $x_i o_p = \sum_{t=1}^M o_{pt} x_{it}$, provides the change to approximate the high-order U-statistics, since the estimation of higher-order statistics requires a tuple of samples. Section 2.4 of the main body provides the theoretical analysis. It reveals that **the bilinear function with multi-head inner-products possesses the property of universal approximation** according to Theorems 2.1 and 2.6. This theoretical result is not trivial. It is obtained via the following steps.
>
> - (Theorem 2.1) Explicitly forming bilinear features (outer-product entries) after a global invertible linear mixing and feeding them into an expressive MLP yields a class of architectures that is universal for continuous high-order dependence mappings on compact domains.
> - (Theorem 2.6) Using multiple inner-product channels (multi-head inner-products) can reconstruct the full outer-product when the number of channels reaches $m^2$ and the projections are chosen appropriately; hence, multi-head constructions can recover universality.
>
> Therefore, we believe the proposed ``parametric high-order dependence measure’' is novel and significant. Your misunderstanding can be attributed to the misinterpretation of the role of the trainable matrix $O$. We hope you find this explanation satisfactory.

---

> > ### Comment · Reviewer_iGCz · 2025-11-28
> >
> > Thank you for your thoughtful and detailed rebuttal. The theoretical justification via Theorems 2.1 and 2.6, establishing universal approximation of high-order dependence through multi-head bilinear forms, now appears sound and nontrivial.
> > Thus, I retract my earlier claim that the method is merely a rebranding of linear projection, the connection to U-statistics via sample tuple combinations, as illustrated in Figure 2(d), is compelling.

---

> ### Author Response · Authors · 2025-11-21
> **Rebuttal part 2**
>
> > Q2. The paper conflates nonlinear correlation (e.g., dCor, HSIC) with low-order and rebrands a bilinear form as high-order, which is technically inaccurate and misleading.
>
> R2.  We want to clarify that the proposed bilinear function approximation to high-oder  U-statistics is accurate with rigorous theoretical analysis. Please refer to Section 2.4 of the main body for the theoretical analysis. It reveals that **the bilinear function with multi-head inner-products possesses the property of universal approximation to high-order U-statistics** according to Theorems 2.1 and 2.6.
>
> Besides, we want to clarify that, similar to U-statistics over M-tuples (M >=3), the proposed bilinear function approximation also explicitly utilizes M-tuple samples. This is based on the recognition that $x_{it}$’s are the samples of random variables $v_i$ of the brain region $I$. Thus, $x_i$ is the collection of these samples, and $ x_iJ$ or $ x_iO$ is the combination of a tuple of samples. Figure 2(d) provides a visual and intuitive case for the 3-order dependence measures, where a 3-tuple of samples, i.e., $(x_{it},x_{is},x_{iq})$ and $(x_{jt},x_{js},x_{jq})$, is use.
>
> Furthermore, the computation cost reduction stems from the reduction of the number of sample tuples and a learnable combination function. Classic high-order U-statistics require enumerating all M-tuples of samples, and lead to high computation cost. The proposed bilinear function approximation *learns to combine* a limited number of sample tuples in an efficient manner. We believe that this paradigm is novel.
>
> In summary, the proposed bilinear function **universally** and **efficiently** approximates high-order U-statistics using M-tuple samples.
>
>
>
>
> > Q3. The paper emphasizes interpretability and clinical relevance, yet provides no evidence of practical clinical value.
>
> R3. Our goal in this work is not to propose novel neurological biomarkers, but to develop a more reliable framework for brain network representation learning. The main contribution lies in the proposed graph-structured estimation approach for learning high-quality brain representations, rather than in conducting a neuroscience discovery study. From this perspective, reproducing well-established ASD-related connectivity patterns should be regarded as important validity evidence that the model captures clinically plausible effects, rather than as a weakness.
>
> Moreover, the interpretability in BRep is not based on post-hoc visualization: the connectivity patterns shown in Fig. 3 and Fig. 11 are intrinsic outputs of the parametric high-order dependence module learned during training. The differential connectivity maps thus reflect patterns captured by the model itself, rather than manually added explanations. We agree that demonstrating prospective clinical impact is beyond the scope of this paper and will soften the wording around “clinical relevance” to emphasize neuroscientific plausibility and interpretability of the learned representations.
>
> > Q4. No analysis of motion artifacts, scanner differences, or missing ROIs common in real clinical fMRI.
>
> R4. The ABIDE and ADHD-200 datasets have undergone rigorous preprocessing, including correction for motion artifacts, scanner variability, and missing ROIs. These procedures are not the focus of our study. Details of the ABIDE preprocessing pipeline can be found at the URL [International Neuroimaging Data-sharing Initiative](https://fcp-indi.github.io/)，[ABIDE Preprocessed](https://preprocessed-connectomes-project.org/abide/). Details of the ADHD-200 preprocessing pipeline can be found at the URL [ADHD-200 Preprocessed](https://preprocessed-connectomes-project.org/adhd200/)，[Preprocessed Connectomes Project](https://preprocessed-connectomes-project.org/)，[preprocessed-connectomes-project/adhd200_athena_scripts: Scripts that implement The Athena Pipeline for the ADHD-200 preprocessed initiative](https://github.com/preprocessed-connectomes-project/adhd200_athena_scripts).
>
> > Q5. ABIDE contains 17 heterogeneous sites, yet the paper uses random splits without site-stratification. Performance may collapse under realistic multi-center conditions.
>
> R5. We would like to clarify that ABIDE is not split by simple random partitioning. In our experiments, we use a site-stratified split, where the proportion of subjects from each site is preserved across the train/validation/test sets. This keeps the site composition of each subset close to that of the full dataset and helps mitigate bias under multi-center conditions.

---

> > ### Comment · Reviewer_iGCz · 2025-11-28
> >
> > I am satisfied with your clarification that the bilinear formulation is not conflating nonlinearity with statistical order, but rather approximating true high-order dependencies in a computationally tractable way. The insight that O enables efficient learning over M-tuples is a meaningful algorithmic advance. Additionally, your point about interpretability being intrinsic to the learned connectivity (rather than post-hoc) is well-argued. Reproducing known neurobiological patterns, in this context, serves as strong validation of the representation quality, which aligns with the paper’s stated goal.

---

> ### Author Response · Authors · 2025-11-21
> **Rebuttal par 3**
>
> > Q6. The model reports population-level ACC/AUC, but clinicians need calibrated, individualized risk scores with uncertainty quantification absent here.
>
> R6. According to your suggestion, we have performed a risk calibration analysis to examine whether the model’s predicted probabilities are reliable at the individual level. Specifically, we plotted the reliability diagram using the model’s predicted confidence scores and computed the Expected Calibration Error (ECE). As shown in Fig. 18, the calibration curve closely aligns with the ideal identity line across the entire confidence range, indicating that the predicted probabilities accurately reflect the true positive rates. On the ABIDE, the model achieves ECE of 0.037, demonstrating excellent calibration quality and confirming that the predicted risk scores faithfully capture each subject’s actual disease likelihood.
>
>
>
> > Q7. The term "graph-structured representation learning'' is presented as novel, but similar ideas appear in BQN (Yang et al., ICML 2025) and FBNetGen (Kan et al., MIDL 2022a).
>
> R7. **While these works do include learnable connectivity components, they differ from our setting in where the graph comes from and how it is learned.** In BQN, the graph is built from a precomputed Pearson-correlation FC matrix, and the subsequent message passing operates on this fixed hand-crafted graph rather than learning FC directly from raw time series. FBNetGen does generate an adjacency matrix from time series, but its graph-generation module is not trained in the same fully end-to-end “time series → learned connectivity → prediction” manner as in our framework.
>
> **By contrast, BRep takes time series as input and explicitly parameterizes a transformation matrix $\mathbf{O}$ from which functional connectivity is constructed and optimized jointly with the prediction objective**. This enables end-to-end learning of FC directly from time series and allows the learned structure to be plugged into different predictors (as shown in Tables 1–2), which we consider a key conceptual and methodological advance over existing learnable-connectivity approaches such as BQN and FBNetGen.

---

> > ### Comment · Reviewer_iGCz · 2025-11-28
> >
> > The inclusion of site-stratified splitting and the new calibration analysis (Fig. 18, ECE = 0.037) directly addresses my concerns about real-world applicability and clinical utility. Excellent calibration at the individual level significantly strengthens the practical relevance of the risk scores.

---

> > ### Comment · Reviewer_iGCz · 2025-11-28
> >
> > Thank you for your thoughtful and detailed rebuttal. I appreciate the care and rigor with which you addressed the concerns raised in my review. Your clarifications have substantially improved my understanding of the method, and I acknowledge that several of my initial critiques were based on a misinterpretation of the role of the trainable matrix O and the theoretical underpinnings of your framework.

---

### Official Review · Reviewer_3u66 · 2025-11-01

**Soundness:** 3
**Presentation:** 3
**Contribution:** 3
**Rating:** 8
**Confidence:** 4

**Summary:**

This paper proposes BRep, a learnable high-order dependence estimator for constructing functional brain networks from BOLD fMRI time series. The key idea is to replace simple (non-)linear correlations with a learnable parametric estimator that can capture higher-order dependencies between the regions. Experiments on ABIDE and ADHD200 show gains over conventional baselines and also demonstrate improvements on GNN/GT/NN-based models as a plug-in HDM module.

**Strengths:**

- The motivation of the paper addresses an important issue in the field (assumptions of connectivity as simple correlations), and neatly formulates this issue as a research problem.
- The method is validated with extensive experiments, including performance benchmarks and neuroscientific interpretation.
- The method is theoretically explained throughout.

**Weaknesses:**

- While I do not see a major/general weakness, please refer to the Questions section for detailed comments.

**Questions:**

### Major
- I agree with the authors' statement (Fig. 1) that a well-defined high-order correlation connectivities require simpler predictors. Currently, the simple predictor is an MLP, which can still be optimized to learn complex relationships. Could the authors provide experimental results on linear approximators on $\mathbf{A}$?
- For the analysis of the optimality of the dimension $D$ in Section 3.2., it would be more convincing if the optimal dimension follows the timeseries length for other data with longer timepoints. Could the authors check if the trend in Fig. 5 follows the timeseries length for data with longer timepoints, or remains optimal at $D=100$?
- Providing qualitative heatmap plots of the learned $\mathbf{A}$ matrix would be helpful to the readers.
- It seems that the trainable parameters within $\mathbf{O}$ include TopK within its backpropagating path, which is not explicitly differentiable. Please clarify.

### Minor
- Some abbreviations are defined multiple times throughout the paper.

---

> ### Author Response · Authors · 2025-11-21
> **Rebuttal**
>
> > Q1. Could the authors provide experimental results on linear approximators on $A$?
>
> Following your suggestion, we have included the experimental results of MLP on $A$, which is now provided in Fig. 17. As shown in the figure, the performance on both ABIDE and ADHD-200 improves when increasing the MLP depth from 1 to 2 layers, and remains relatively stable between 2 and 3 layers. This observation indicates that the representations learned by HDM are highly expressive, such that even simple predictors can achieve strong performance.
>
> > Q2. Could the authors check if the trend in Fig. 5 follows the timeseries length for data with longer timepoints, or remains optimal at $D=100$ ?
>
> R2. Due to practical limitations in clinical imaging protocols, fMRI datasets with substantially longer time series are not yet common. Nevertheless, we plan to investigate more datasets and perform additional evaluations in future work.
>
>
>
> > Q3. Providing qualitative heatmap plots of the learned matrix would be helpful to the readers.
>
> R3. According to your advice, we have added the following heatmap of $A$​ into the revised manuscript  as shown in Fig. 15. In the revised manuscript, the connectivity heatmaps (for NC and ASD templates and ASD–NC difference) of BRep and VGAE are reported in Figs. 15 and 16, **where all ROIs are reordered into six atlas-based macro–anatomical groups—Prefrontal (PFR), Frontal (FR), Parietal (PR), Occipital (OR), Temporal (TR), and Subcortical (SUB)**. The analyses are as follows.
>
> **The BRep-based NC and ASD templates (Fig. 15(a,b)) show a recognizable modular structure: within-block connectivity (block diagonals) is stronger than between-block connectivity** [1]. The NC and ASD templates remain visually similar, which is consistent with ASD being a neurodevelopmental condition with subtle, distributed FC alterations rather than gross disruption of whole networks [2]. Note that the BRep difference map (Fig. 15(c)) shows spatially coherent clusters of altered connectivity, mainly in TR–FR/PFR, TR–OR, TR–PR, and SUB–PFR connections, corresponding to atypical coupling between sensory, visual, social–cognitive, and subcortical–prefrontal systems that have frequently been reported in ASD studies [3-6].
>
>
>
>
>
> > Q4. It seems that the trainable parameters within include TopK within its backpropagating path, which is not explicitly differentiable.
>
> R4. The TopK module in our implementation is differentiable. This is implemented via PyTorch’s torch.topk, whose gradients are handled by the autograd engine.
>
>
>
> > Q5. Some abbreviations are defined multiple times throughout the paper.
>
> R5. We have revised the manuscript to remove repeated definitions.
>
>
>
> [1] Power, J. D., Cohen, A. L., Nelson, S. M., Wig, G. S., Barnes, K. A., Church, J. A., Vogel, A. C., Laumann, T. O., Miezin, F. M., & Schlaggar, B. L. (2011). Functional network organization of the human brain. *Neuron*, *72*(4), 665-678.
>
> [2] Holiga, Š., Hipp, J. F., Chatham, C. H., Garces, P., Spooren, W., D’Ardhuy, X. L., Bertolino, A., Bouquet, C., Buitelaar, J. K., & Bours, C. (2019). Patients with autism spectrum disorders display reproducible functional connectivity alterations. *Science Translational Medicine*, *11*(481), eaat9223.
>
> [3] Supekar, K., Uddin, L. Q., Khouzam, A., Phillips, J., Gaillard, W. D., Kenworthy, L. E., Yerys, B. E., Vaidya, C. J., & Menon, V. (2013). Brain hyperconnectivity in children with autism and its links to social deficits. *Cell reports*, *5*(3), 738-747.
>
> [4] Long, Z., Duan, X., Mantini, D., & Chen, H. (2016). Alteration of functional connectivity in autism spectrum disorder: effect of age and anatomical distance. *Scientific reports*, *6*(1), 26527.
>
> [5] Uddin, L. Q., Supekar, K., & Menon, V. (2013). Reconceptualizing functional brain connectivity in autism from a developmental perspective. *Frontiers in human neuroscience*, *7*, 458.
>
> [6] Woodward, N. D., Giraldo-Chica, M., Rogers, B., & Cascio, C. J. (2017). Thalamocortical dysconnectivity in autism spectrum disorder: An analysis of the Autism Brain Imaging Data Exchange. *Biological Psychiatry: Cognitive Neuroscience and Neuroimaging*, *2*(1), 76-84.

---

> > ### Comment · Reviewer_3u66 · 2025-11-28
> >
> > I appreciate the authors' response. Does [1-layer MLP] in Q1.R1 refer to an MLP with one hidden layer? If so, I would like to remind the authors that the question was intended to check if a single linear layer without nonlinearity suffices (as in linear probing). I am not expecting that the performance can match MLPs, but I would be more convinced if BRep provides an advantage over conventional correlation connectivities in a linear probing setting.

---

> > > ### Author Response · Authors · 2025-11-29
> > > **Rebuttal**
> > >
> > > You are right that our previous "1-layer MLP" still included a nonlinearity. To test the performance benefits of BRep across both linear and nonlinear settings, we evaluate four combinations: (i) conventional Pearson correlation connectivities with a linear probe, (ii) Pearson with a nonlinear MLP, (iii) BRep connectivities with a linear probe, and (iv) BRep (with a nonlinear MLP). The new results in Fig. 17 show that, in the linear probing setting, BRep consistently provides an advantage over conventional correlation connectivities on both ABIDE and ADHD-200, while its performance does not match that of the nonlinear MLPs. This indicates that the gain of BRep comes from the learned BRep-based connectivity itself rather than merely from using a more expressive predictor.

---

### Official Review · Reviewer_FT4h · 2025-11-01

**Soundness:** 3
**Presentation:** 2
**Contribution:** 2
**Rating:** 4
**Confidence:** 3

**Summary:**

This paper reframes brain functional connectivity as a **learnable, graph-structured representation** and proposes BRep, which replaces fixed correlation graphs with a **parametric high-order dependence estimator** trained end-to-end. Concretely, BOLD time series are mapped via a learnable matrix \(O\) to form \(Z=\hat{X}O\), and the adjacency is built as \(A=\sigma(\mathrm{Norm}(\mathrm{TopK}(ZZ^\top)))\). Across ABIDE and ADHD-200, BRep paired with simple MLP/BQN heads **outperforms or matches** GNN and Graph Transformer baselines; inserting the proposed HDM as a plug-in also improves existing models. The method further employs a denoising regularizer and reports sensitivity to the HDM dimension.

**Strengths:**

- **Conceptually clear reformulation.** Treating the connectivity graph itself as the representation, rather than a fixed input to a downstream GNN, motivates an end-to-end pipeline that simplifies the predictor without sacrificing accuracy.
- **Unified, parametric high-order dependence (HDM).** The paper connects linear/nonlinear correlations and proposes \(r^{[\mathrm{high}]}_{ij} = (x_i O)(x_j O)^\top\) as a learnable surrogate, followed by principled sparsification (TopK) and normalization.
- **Consistent empirical gains and plug-in utility.** BRep attains strong results on ABIDE/ADHD-200 and yields additional improvements when HDM is inserted into GCN/BrainNETTF, indicating broad applicability.

**Weaknesses:**

1. **Insufficient theoretical grounding of the HDM surrogate.**
   The approximation \(JJ^\top \approx OO^\top\) and the use of \( (x_i O)(x_j O)^\top \) as a proxy for high-order dependence are motivated heuristically. Formal properties (e.g., bias/consistency under realistic noise models, identifiability or approximation guarantees) are not established, leaving it unclear when the surrogate faithfully captures higher-order interactions.

2. **Limited evaluation under site variability and distribution shift.**
   ABIDE/ADHD-200 are multi-site, yet no site-held-out or scanner-held-out protocol is reported. Without cross-site/cross-dataset tests, robustness to dataset shift and potential site leakage remain unassessed.

3. **Attribution of gains among HDM, TopK, and denoising is unclear.**
   Ablations remove normalization/denoising, but the specific contribution of HDM vs. TopK(\(ZZ^\top\)) vs. the GCN-based denoising regularizer is not disentangled. Sensitivity to \(k\), \(\lambda\) (denoising weight), and the activation \(\sigma\) needs to be characterized to support the claimed mechanisms.

4. **Scalability and compute reporting are incomplete.**
   With \(O\in\mathbb{R}^{D\times D}\) and the construction of \(ZZ^\top\), memory/time complexity and wall-clock performance are not reported as functions of \(N\) (ROIs) and \(D\) (time length). Practical feasibility on higher-resolution atlases or longer scans is thus unclear.

5. **Generalization across variable time-series lengths is not addressed.**
   Results suggest \(D{=}100\) performs best because \(O\) is square. However, the behavior under variable \(D\) (different TRs, windowing, missing timepoints) is not evaluated, despite being common in real datasets.

6. **Interpretability evidence lacks statistical rigor.**
   Differential ASD–NC connectivity visualizations are suggestive but lack group-level statistical testing (e.g., permutation tests, FDR control) and site-stratified analyses, making the biological claims preliminary.

### Minor
- **Typos/wording:** “multi-layer **perception** (MLP)” → *perceptron*; “HSCI” → *HSIC*; “**caculated**” → *calculated*; “**deontes**” → *denotes*.
- **Clarity:** In Eq. (6), restate shapes for \(X,\hat{X},Z,O\) and \(A\in\mathbb{R}^{N\times N}\).
- **Protocol:** Specify whether splits are **site-stratified** and how repeated sessions per subject (if any) are handled to avoid leakage.

**Questions:**

1. **Theory:** Under what assumptions is the HDM estimator \( (x_i O)(x_j O)^\top \) a consistent or bounded-bias surrogate for high-order dependence? Can you provide error bounds or identifiability conditions (e.g., under sub-Gaussian noise or bounded moments)?
2. **Distribution shift:** Do you have **site-held-out** evaluations (leave-one-site-out/leave-k-sites-out) and **cross-dataset transfer** (e.g., pretrain on ABIDE, fine-tune/evaluate on ADHD-200)? These would clarify robustness to scanner/protocol shift.
3. **Ablation attribution:** Can you isolate contributions via: (i) HDM only (no denoising; fixed \(k\)), (ii) denoising only (Pearson graph), (iii) HDM+denoising; and provide sensitivity to \(k\in\{20,40,80,160\}\), \(\lambda\), and \(\sigma\) on both datasets?
4. **Scalability:** Please report parameter counts, peak GPU memory, and wall-clock for representative \(N\in\{200,400\}\) and \(D\in\{100,200,400\}\). How is \(ZZ^\top\) batched to avoid \(O(N^2D)\) bottlenecks in practice?
5. **Variable-length handling:** If time lengths differ across subjects, do you window, resample, or mask? Can you show robustness to time-window length and stride to support the \(D{=}100\) choice beyond the square-\(O\) argument?
6. **Interpretability statistics:** For ASD–NC differentials, can you include edge-wise permutation tests with FDR control and site-stratified analyses to rule out site confounds?

---

> ### Author Response · Authors · 2025-11-21
> **Rebuttal part 1**
>
> > **W1&Q1**. Insufficient theoretical grounding of the HDM surrogate.
>
> R1. Thanks for your professional comments. Based on your suggestion, we further enhance the theoretical contributions by providing the universal approximation theorem of bilinear function, i.e., $((x_i O)(x_j O)^\top)$ toward high-order dependence measures.
>
> Please refer to Section 2.4 of the main body. It reveals that **the bilinear function with multi-head inner-products possesses the property of universal approximation** according to Theorems 2.1 and 2.6. We hope you find this theoretical result satisfactory.
>
> We would also like you to note that this theoretical result is not trivial. It is obtained via the following three steps.
>
> - (Theorem 2.1) Explicitly forming bilinear features (outer-product entries) after a global invertible linear mixing and feeding them into an expressive MLP yields a class of architectures that is universal for continuous high-order dependence mappings on compact domains.
>
> - (Theorem 2.4) Reducing the relation between two sample groups to a single scalar inner-product and applying a single-variable nonlinearity is strictly limited in expressivity and cannot approximate arbitrary continuous high-order dependence functions.
>
> - (Theorem 2.6) Using multiple inner-product channels (multi-head inner-products) can reconstruct the full outer-product when the number of channels reaches $m^2$ and the projections are chosen appropriately; hence, multi-head constructions can recover universality.
>
> > **W2&Q2**. Limited evaluation under site variability and distribution shift. ABIDE/ADHD-200 are multi-site, yet no site-held-out or scanner-held-out protocol is reported.
>
> R2. **Site variability was taken into account during dataset partitioning**. For ABIDE, a site-stratified split was performed based on the number of samples at each site, so that the train/validation/test sets preserve the original site-wise distribution as closely as possible. For ADHD-200, because explicit site information is not available in our processed version of the dataset following [1, 2], a class-stratified split was adopted, which effectively prevents class imbalance while maintaining the overall label distribution.
>
> Distribution shift is indeed an important point, especially in the context of brain network analysis. However, cross-site or cross-dataset generalization tests address a different research question and were therefore not adopted as the primary focus of the present study. We appreciate the reviewer’s suggestion, and more comprehensive distribution-shift evaluations will be considered in future work.
>
> [1] Peng, C., Huang, Y., Dong, Q., Yu, S., Xia, F., Zhang, C., & Jin, Y. (2025). Biologically plausible brain graph transformer. *arXiv preprint arXiv:2502.08958*.
>
> [2] Yang, L., Liu, Y., Zhuo, J., Jin, D., Wang, C., Wang, Z., & Cao, X. Do We Really Need Message Passing in Brain Network Modeling?. In *Forty-second International Conference on Machine Learning*.
>
> > **W3&Q3**. Attribution of gains among HDM, TopK, and denoising is unclear.
> > Sensitivity to (k), (\lambda) (denoising weight), and the activation (\sigma) needs to be characterized to support the claimed mechanisms.
>
> R3. In the submitted manuscript, the impact of the mentioned hyperparameters were analyzed in Section 3.3 (Additional Analysis — Hyperparameter Analysis). The corresponding results of the number of HDM layers, the TopK connection, the denoising loss weight, and the noise ratio are provided in Fig. 7, Fig. 8, Fig. 9, a.nd Fig. 10, respectively.

---

> ### Author Response · Authors · 2025-11-21
> **Rebuttal part 2**
>
> > **W4&Q4**. Scalability and compute reporting are incomplete.
>
> Q4. The time complexity of the proposed method is $O(N^2D+ND^2)$, and the space complexity is $O(N^2+ND)$,where $N$ denotes the number of ROIs, $D$ denotes time series length. The runtime of the model on different datasets is summarized in the table below. All results were obtained under the same experimental setting, corresponding to 200 training epochs, followed by validation, testing, and metric computation.
>
> | dataset  | running time(s) |
> | :------: | :-------------: |
> |  ABIDE   |      37.27      |
> | ADHD-200 |      18.89      |
> |   ADNI   |      8.95       |
>
> In this implementation, o special handling is applied to the computation of $ZZ^\top$. The brain atlases commonly used in functional connectivity analysis are relatively modest in size (e.g., Craddock-200 with 200 ROIs, AAL with 116 ROIs). Therefore, both the time and memory cost of directly computing $ZZ^\top$ remain well within a practical and acceptable range. For larger-scale datasets with substantially more ROIs, a blockwise or batched computation strategy can be adopted to further reduce the computational overhead, and we will clarify this point in the revised manuscript.
>
>
>
> > **W5&Q5**. Generalization across variable time-series lengths is not addressed.
>
> R5. In practical neuroimaging preprocessing workflows, variability in scan length (e.g., different TRs and numbers of timepoints) is typically handled at the preprocessing stage by standardizing each subject’s time series to a consistent temporal length via truncation, padding, or fixed-length windowing. This constructs an aligned temporal dimension for downstream modeling. Consequently, under standard preprocessing pipelines, the input time-series length D is kept consistent across subjects, and the use of a fixed D in our study is consistent with common data-processing practice. The specific pre-processing process can be found in the URL. [ABIDE Preprocessed](https://preprocessed-connectomes-project.org/abide/), [ADHD-200 Preprocessed](https://preprocessed-connectomes-project.org/adhd200/)
>
>
>
> > **W6&Q6**. Interpretability evidence lacks statistical rigor.
>
> R6. According to your suggestion, we have performed edge-wise permutation testing (2,000 permutations) on the 200×200 functional connectivity matrices for ASD (n = 493) and NC (n = 516), followed by Benjamini–Hochberg FDR correction (q < 0.05), as shown in Fig. 11. Only connections that remained significant after FDR correction were retained. The resulting set of significant edges shows strong overlap with those exhibiting the largest raw ASD–NC differences, indicating that the model-identified group differences are stable, reliable, and unlikely to be driven by noise. Furthermore, the perturbed top-20 difference matrix closely matches the original top-20 difference matrix, demonstrating that the observed patterns are robust to random fluctuations.
>
>
>
> > Q7. Minor mistakes
>
> R7. Thank you for careful check. All spelling errors have been corrected in the revised version, and the shapes of $\mathbf{X}, \mathbf{\hat{X}}, \mathbf{Z} \in \mathbb{R}^{N\times D}, \mathbf{A} \in \mathbb{R}^{N\times N}, \mathbf{O} \in \mathbb{R}
> ^{D\times D}$ have been explicitly added in Section 2.3 for clarity.

---

### Meta-Review · Area_Chair_AzH5 · 2026-01-17

**Summary:**

Overall, this manuscript is borderline with respect to the pile.

Reviewers note their confusion about higher order dependence measures; this is perhaps due to the manuscript's writing and communication of these ideas, not due to an incorrect technical definition. The authors' proposed method does indeed use n-tuples of arbitrary size (though in implementation there is a selection mechanism limiting overfit potential.

However, concerns about generalization are not addressed: windowing is required (to a fixed context window that matches the ABIDE dataset), and transference of inference results to other datasets or other, unseen sites is explicitly avoided; the experimental design specifically stratifies on site to _mix_ sites, not induce holdout sites. This is exactly the opposite of the "real-world" situation, which in itself is already contrived relative to the scientific or imagined clinical use case, which for clinical work is only beginning to emerge in a few pathologies.

**Reviewer Concerns:**

Interpretation, generalization (`FT4h` and `iGCz`), site-wise variance/overfitting (`iGCz`), for which `iGCz` is apparently satisfied by the response, but I am not..

Approximation convergence rates the O-matrix estimator (`FT4h`).

To be clear, the authors _did_ address many of the theoretical/methodological concerns or misunderstandings, and this taken into account makes the paper near the boundary.

**Reviewer Scores:**

It is clear multiple very low scores would increase (2->higher, 6).

It is unclear if this pushes every score, and it is doubtful that scores would reach the clear accept (8) range.

---

### Decision · Program_Chairs · 2026-01-26

Reject